



# Large contribution of meteorological factors to inter-decadal changes in regional aerosol optical depth

Huizheng Che[1*], Ke Gui[1,2], Xiangao Xia[3,2], Yaqiang Wang[1], Brent N. Holben[4], Philippe Goloub[5],

Emilio Cuevas-Agulló[6], Hong Wang[1], Yu Zheng[7,1], Hujia Zhao[1], Xiaoye Zhang[1*]

State Key Laboratory of Severe Weather (LASW) and Key Laboratory of Atmospheric
    Chemistry (LAC), Chinese Academy of Meteorological Sciences, CMA, Beijing, 100081,
    China

College of Earth and Planetary Sciences, University of Chinese Academy of Sciences,
    Beijing, 100049, China

Key Laboratory for Middle Atmosphere and Global Environment Observation (LAGEO),
    Institute of Atmospheric Physics, Chinese Academy of Sciences, Beijing, 100029, China

NASA Goddard Space Flight Center, Greenbelt, MD, USA
Laboratoire d'Optique Amosphérique, Université des Sciences et Technologies de Lille,
    59655, Villeneuve d'Ascq, France

Centro de Investigación Atmosférica de Izaña, AEMET, 38001 Santa Cruz de Tenerife , Spain
Collaborative Innovation Center on Forecast and Evaluation of Meteorological Disasters,
    Nanjing University of Information Science & Technology, Nanjing 210044, China

***Correspondence to***: Huizheng Che (chehz@cma.gov.cn) & Xiaoye Zhang (xiaoye@cma.gov.cn)



## Abstract

Aerosol optical depth (AOD) has become a crucial metric for assessing global climate change. Although global and regional AOD trends have been studied extensively, it remains unclear what factors are driving the inter-decadal variations in regional AOD and how to quantify the relative contribution of each dominant factor. This study used a long-term (1980–2016) aerosol dataset from the Modern-Era Retrospective Analysis for Research and Applications, version 2 (MERRA-2) reanalysis, along with two satellite-based AOD datasets (MODIS/Terra and MISR) from 2001 to 2016, to investigate the long-term trends in global and regional aerosol loading. Statistical models based on emission factors and meteorological parameters were developed to identify the main factors driving the inter-decadal changes of regional AOD and to quantify their contribution. Evaluation of the MERRA-2 AOD with the combined in-situ measurements of AERONET and the China Aerosol Remote Sensing Network indicated significant spatial agreement on the global scale ($r = 0.84$, RMSE = 0.14, and MAE = 0.07). In general, MERRA-2 was able to quantitatively reproduce the annual and seasonal AOD trends on both regional and global scales, as observed by MODIS/Terra, albeit some differences were found when compared to MISR. Over the 37-year period in this study, significant decreasing trends were observed over Europe and the eastern United States. In contrast, eastern China and South Asia showed AOD increases, but the increasing trend of the former reversed sharply in the most recent decade. The statistical analyses suggested that the meteorological parameters explained a larger proportion of the AOD variability (20.4%–72.8%) over almost all regions of interest (ROIs) during 1980–2014 when compared with emission factors (0%–56%). Further analysis also showed that $SO_2$ was the dominant emission factor, explaining 12.7%–32.6 % of the variation in AOD over anthropogenic aerosol–dominant regions, while BC or OC was the leading factor over the biomass burning–dominant (BBD) regions, contributing 24.0%–27.7% of the variation. Additionally, wind speed was found to be the leading meteorological parameter, explaining 11.8%–30.3% of the variance over the mineral dust–dominant regions, while ambient humidity (including soil moisture and relative humidity) was the top meteorological parameter over the BBD regions, accounting for 11.7%–35.5% of the variation. The results of this study indicate that the variation in meteorological parameters is a key factor in determining the inter-decadal change in regional AOD.

## 1. Introduction

Atmospheric aerosols play a key role in the energy budget of the Earth's climate system through aerosol–radiation interactions (direct effect) and aerosol–cloud interactions (indirect effect). On the one hand, by absorbing and scattering solar and terrestrial radiation, aerosols generally cool the Earth's surface and heat the atmosphere, depending on the absorption level of the aerosols (McCormick and Ludwig 1967; Ding et al., 2016; Sun et al., 2018; Zheng et al., 2019). This effect is termed the aerosol direct effect. The cooling effect of aerosols may partly counteract the warming caused by the



increase in $CO_2$ and other greenhouse gases in the past several decades (IPCC, 2007).
On the other hand, by acting as cloud condensation nuclei or ice nuclei, not only can
aerosols alter the microphysical and radiative properties of clouds, as well as their
lifetimes (Rosenfeld et al., 2019; Andreae 2009), but they can also change the
precipitation efficiency [depending on the aerosol type (Jiang et al., 2018)], modify the
characteristics of the atmospheric circulation, and affect the global hydrological cycle
(Ramanathan et al., 2001; Ackerman et al., 2000; Hansen et al., 1997; Sarangi et al.,
2018). This effect is termed the aerosol indirect effect. Furthermore, depending on their
physical and chemical properties, as well as their composition, aerosols can affect
ecosystems (Yue et al., 2017; Liu et al., 2017), atmospheric visibility (Che et al., 2007;
Wang et al., 2009; Che et al., 2014), and even human health [such as through their roles
in lung cancer, respiratory infection, and cardiovascular disease (Silva et al., 2013;
Lelieveld et al., 2015; Cohen et al., 2017)]. Unlike the long-lived greenhouse gases (e.g.,
$CO_2$, $CH_4$ and $N_2O$), aerosols produced via anthropogenic activity or naturally have
relatively short life spans and large spatial and temporal variability. Therefore, it is
essential to investigate the long-term variability and inter-decadal trends of atmospheric
aerosol loadings on both regional and global scales.
Aerosol optical depth (AOD), representing the attenuation of sunlight induced by
aerosols and serving as an important measure of aerosol loading, has become a crucial
metric in assessing global climate change and the effects of aerosols on radiation,
precipitation and clouds. Through the efforts of scientists in various countries over the
past three decades, a series of AOD datasets with different time spans derived from
continuous ground-based and satellite observations have been accumulated. These
datasets have been widely employed to investigate the long-term annual and seasonal
trends of AOD at global and regional scales. Although ground-based observations have
limited spatial and/or temporal coverage, they can provide more detailed information on
aerosol properties and long-term variations for satellite and model validation. For
example, using the long-term and high-quality AOD datasets from the Aerosol Robotic
Network (AERONET), Li et al. (2014) found that North America and Europe
experienced a uniform decrease in AOD from 2000 to 2013. Che et al. (2015) estimated
the change in AOD based on AOD data at 12 long-term ground-based sites in China
from the China Aerosol Remote Sensing Network (CARSNET) and found that AOD
showed a downward trend from 2006 to 2009 and an upward trend from 2009 to 2013.
Compared with the spatial sparseness of ground-based observations, inferences from
satellite-based sensors can provide a global perspective of AOD change, due to their
continuous spatial measurements. Previous studies (Hsu et al., 2012; Pozzer et al., 2015;
Mehta et al., 2016; Klingmüller et al., 2016; De Leeuw et al., 2018; Zhang and Reid
2010) have investigated global and regional AOD trends by using multiple satellite
observations, including the Moderate Resolution Imaging Spectroradiometer (MODIS),
Multiangle Imaging Spectroradiometer (MISR), the Sea-viewing Wide Field-of-view
Sensor (SeaWiFS), and others. These studies have shown increased AODs over eastern
China, India, the Middle East (ME), and the Bay of Bengal, and decreased AODs over
the eastern United States (EUS) and Europe.
In general, regional AOD changes are closely linked to the variations in natural



emissions driven by meteorological conditions (such as mineral dust) and local anthropogenic emissions associated with economic and population growth. For example, over anthropogenic aerosol–dominant regions, most of the primary pollutant emissions [such as black carbon (BC)] and aerosol precursors (such as $SO_2$, $NO_x$ and $NH_3$) in North America and Europe have declined in response to emissions control (Hammer et al., 2018). In contrast, pollutant emissions and their precursors in the rapidly developing countries (such as India and China) have increased over the past few decades, attributable to enhanced industrial activity. However, as a consequence of clean-air actions, anthropogenic emissions in China have declined significantly in recent years (Zheng et al., 2018). It has been proven that these changes in local pollutant emissions or aerosol precursors over the above regions can to a certain extent explain the regional AOD variability, as observed in long-term satellite aerosol data records (Meij et al., 2012; Itahashi et al., 2012; Feng et al., 2018). On the other hand, various studies have shown that meteorological changes play a major role in determining the inter-decadal trend of AOD over mineral dust–dominant regions, particularly in the Sahara Desert (SD) and the ME (Pozzer et al., 2015; Klingmüller et al., 2016). Based on model simulations during 2001–2010, Pozzer et al. (2015) suggested that, over biomass burning–dominant regions, the changes in both meteorology and emissions are equally important for driving AOD trends. Considering the localized changes in anthropogenic aerosol emissions and meteorological conditions in different regions, a key question is whether these factors are responsible for the regional AOD trends, or which main factors dominate the trends. Therefore, it is important to investigate the cause of regional AOD trends in terms of the variations in both anthropogenic emissions and meteorological factors for projecting the response of the earth system to future changes.

In this study, we used a long-term (1980–2016) aerosol dataset obtained from the Modern-Era Retrospective Analysis for Research and Applications, version 2 (MERRA-2) reanalysis, along with two satellite-based datasets (MODIS/Terra and MISR) during 2001–2016, to conduct a comprehensive estimation of global and regional AOD trends over different periods. To ensure the reliability of the trend assessment, 468 AERONET sites and 37 CARSNET sites with continuous observations for at least one year were used to assess the performance of the MERRA-2 AOD on a global scale. Twelve regions dominated by different aerosol types were selected to explore the relationships between local anthropogenic emissions, meteorological factors, and regional AOD. Furthermore, stepwise multiple linear regression (MLR) models were developed to estimate the regional AOD as a function of emission factors and other meteorological parameters, which allowed the influences of emissions and meteorology to be separated. Then, the Lindeman, Merenda and Gold (LMG) method was applied to the MLR models to identify the main factors driving the regional AOD variability and to quantitatively evaluate the contribution of each driving factor.



## 2. Data and methods

### 2.1 MERRA-2 aerosol reanalysis data

MERRA-2 is the latest atmospheric reanalysis version for the modern satellite era provided by the NASA Global Modeling and Assimilation Office (Gelaro et al., 2017), using the Goddard Earth Observing System, version 5 (GEOS-5), earth system model (Molod et al., 2012, 2015), which includes atmospheric circulation and composition, ocean circulation and land surface processes, and biogeochemistry. Note that, in MERRA-2, in addition to providing assimilation of traditional meteorological observations, a series of AOD observation datasets, including bias-corrected AODs retrieved from the Advanced Very High Resolution Radiometer instrument over the oceans (Heidinger et al., 2014) and MODIS (onboard both the Terra and Aqua satellites) (Levy et al., 2010; Remer et al., 2005), and non-bias-corrected AODs retrieved from MISR (Kahn et al., 2005) over bright surfaces and ground-based AERONET observations (Holben et al., 1998), were also assimilated within the GEOS-5 earth system model. An overview of the MERRA-2 modeling system and a more detailed description of aerosols in the MERRA-2 system can be found in Gelaro et al. (2017) and Buchard et al. (2017), respectively. In this study, the three-hourly instantaneous AOD datasets, at a resolution of 0.5 ° latitude by 0.625 ° longitude, were used for evaluation, while the monthly mean AOD values were used for climate analysis.

### 2.2 Satellite aerosol data

Two AOD datasets during 2001–2016 retrieved from MODIS and MISR, both onboard the Terra platform, were used in this study. The MODIS sensor onboard the Terra satellite observes the Earth at multiple wavelengths (range: 410–1450 nm; 36 bands) with a 2330-km swath, which has provided near-daily global coverage since 2000 (King et al., 2003; Levy et al., 2015). This study employed the combined Dark Target/Deep Blue AOD algorithm at 550 nm, with a 1 ° × 1 ° resolution, from the Level 3 monthly global aerosol dataset for MODIS Terra, Collection 6.1. Note that MODIS/Aqua L3 was not used because it started late (June 2002). In addition, compared with MODIS/Aqua AOD monthly datasets, MODIS/Terra AOD shows similar performance worldwide (Fig. S1). Thus, with its longer observation time, MODIS/Terra was used in this study. The expected errors of the Level 2 MODIS AOD data have been estimated to be about ±(0.03 + 0.05 × AOD) over ocean and ±(0.05 + 0.15 × AOD) over land (Levy et al., 2013).

Total column AOD observations from the MISR sensor onboard the Terra satellite, which provides observations of the Earth and atmosphere with nine different along-track viewing zenith angles at four different spectral bands (440–866 nm) (Diner et al., 1998), were utilized. It should be noted that, although MISR has a much narrower swath (~360 km) compared with MODIS, the multi-angle observation from MISR provides the capability for retrieving a more reliable AOD over bright surfaces such as desert areas (Diner et al., 1998; Kahn et al., 2010). The AOD retrieval in the





555-nm channel from monthly global aerosol datasets at a spatial resolution of 0.5 ° × 0.5 ° were used in this study. The uncertainty of the MISR Level 2 AOD data over land and ocean has been estimated to be ±0.05 or ±(0.2 × AOD) (Kahn et al., 2005). Note that the wavelength of AOD (555 nm) reported by MISR is different from that of the MERRA-2 and MODIS/Terra datasets (550 nm); however, this slight wavelength difference is not expected to affect our analysis and conclusions regarding AOD annual and seasonal trends.

## 2.3  Ground-based reference data: AERONET and CARSNET

Owing to the accuracy of ground-based AOD observations, long-term instantaneous AOD observation records from two independent operational networks—AERONET and the CARSNET—were used to validate the three-hourly MERRA-2 AOD values. Since there are not enough long-term AERONET observations in China, it was necessary to examine the performance of the MERRA-2 analyzed AOD fields using additional AOD observations from CARSNET. CARSNET is a ground-based network for monitoring aerosol optical properties that was first established by the China Meteorological Administration in 2002 (Che et al., 2009). Both AERONET and CARSNET use the same types of sunphotometers, which can observe direct solar and sky radiances at seven wavelengths (typically 340, 380, 440, 500, 670, 870 and 1020 nm) within a 1.2 ° full field of view at intervals of about 15 min (Holben et al., 1998; Che et al., 2009). For CARSNET, operating instruments are calibrated and standardized using CARSNET reference instruments, which in turn are regularly calibrated at Izaña, Tenerife, Spain, together with the AERONET program (Che et al., 2009; Che et al., 2018). The cloud-screened AOD [based on the work of Smirnov et al. (2000)] in CARSNET has the same accuracy as AERONET, with an estimated uncertainty of 0.01–0.02 (Eck et al., 1999; Che et al., 2009).

In this work, we collected ground-based AOD observations (more than one year of data) from 468 AERONET sites worldwide and 37 CARSNET sites in China. The locations of these ground-based sites are shown in Fig. 1. Detailed information about these AERONET and CARSNET sites is given in Tables S4 and S5. The combined instantaneous AOD data collected by AERONET (quality-assured and cloud-screened Level 2.0 data) during 1993–2016 and CARSNET (cloud-screened Level 2.0 data) during 2002–2014 were used. Moreover, to ensure the reliability of AOD evaluation, the AOD measurements in two adjacent channels (i.e., 440 and 675 nm) from AERONET and CARSNET were subsequently interpolated to 550 nm for MERRA-2, using a second-order polynomial fit to ln (AOD) vs. ln (wavelength) (Eck et al., 1999).

## 2.4  Emissions inventory and meteorological data

The anthropogenic emissions inventories used in this study were obtained from the Peking University (PKU) website (http://inventory.pku.edu.cn/), including total suspended particles (TSP) (Huang et al., 2014), $SO_2$ (Su et al., 2011), BC (Wang et al., 2014), and organic carbon (OC) (Huang et al., 2015), with a spatial resolution of 0.1 ° × 0.1 ° and spanning the period 1980–2014. The emissions were calculated using a





bottom-up approach based on fuel consumption and an emissions factor database.
Huang et al. (2015) showed that the PKU emissions inventories are broadly similar to
those of EDGARv4.2 (Edgar, 2011). Monthly meteorological fields from the
MERRA-2 global reanalysis were also utilized, including total surface precipitation,
surface wind speed, surface relative humidity (RH), mean sea level pressure, et. These
data have a spatial resolution of 0.5 $^\circ$ × 0.625 $^\circ$ and span the period 1980–2016 (Gelaro
et al., 2017). For more detailed information on the selected meteorological parameters,
see Table 1.

## 2.5  ROIs

In this study, 12 regions of interest (ROIs) dominated by different aerosol types
were selected to study the long-term trends in regional aerosol loading and how they
are related to local emission changes as well as the variation in meteorological
variables. These 12 ROIs included three mineral dust–dominant regions [SD (17 °W–
20 °E, 3 °N–25 °N), ME (38 °E–56 °E, 14 °N–33 °N), and Northwest China (NWC;
73 °E–94 °E, 35 °N–47 °N)], three biomass burning–dominant regions [the Amazon
Zone (AMZ; 46 °W–60 °W, 1 °S–22 °S), Central Africa (CF; 12 °E–33 °E, 2 °S–18 °S)
and Southeast Asia (SEA; 96 °E–127 °E, 8 °S–18 °N)], and six anthropogenic aerosol–
dominant regions [EUS (73 °W–94 °W, 29 °N–45 °N), western Europe (WEU; 10 °W–
18 °E, 37 °N–59 °N), South Asia (SA; 72 °E–90 °E, 10 °N–30 °N), northern China (NC;
108 °E–120 °E, 30 °N–40 °N), southern China (SC; 108 °E–120 °E, 20 °N–30 °N) and
Northeast Asia (NEA; 125 °E–145 °E, 30 °N–41 °N)]. The geographical boundaries of
these ROIs are shown in Fig. 1.

## 2.6  Statistical analysis

### 2.6.1  Comparison methods

AOD data from the 468 AERONET sites worldwide and the 37 CARSNET sites
in China were used to evaluate the performance of the three-hourly AOD datasets
from MERRA-2. To ensure the accuracy of the assessment, instantaneous
ground-based AOD observations within one hour, obtained from AERONET and
CARSNET, were averaged as the hourly mean AOD and compared with those from
the MERRA-2 three-hourly AOD datasets.
The errors and quality of the MERRA-2 AOD retrievals are reported using the
mean absolute error [MAE, Eq. (1)], root-mean-square error [RMSE, Eq. (2)] and the
relative mean bias [RMB, Eq. (3)]. In addition, linear regression parameters were also
included in this study, including the slope, the *y*-intercept, and correlation coefficient
(*R*).

$$\text{MAE} = \frac{1}{n}\sum_{i=1}^{n}\left|\text{AOD}_{(\text{MERRA-2})i} - \text{AOD}_{(\text{Ground-based})i}\right| \tag{1}$$

$$\text{RMSE} = \sqrt{\frac{1}{n}\sum_{i=1}^{n}\left(\text{AOD}_{(\text{MERRA-2})i} - \text{AOD}_{(\text{Ground-based})i}\right)^2} \tag{2}$$



$$RMB = \overline{\left(AOD_{MERRA\text{-}2}/AOD_{Ground\text{-}based}\right)} \qquad (3)$$

## 2.6.2 Trend analysis and stepwise MLR model

Long-term trend analysis of the AOD from MERRA-2, MODIS/Terra and MISR was performed, on monthly time series data, using ordinary least-squares linear regression—a technique widely employed for trend analysis of aerosol data (Hsu et al., 2012; Pozzer et al., 2015; Klingmüller et al., 2016; Ma et al., 2016; Hammer et al., 2018). Prior to regression, these data were first deseasonalized by subtracting the monthly mean for different study periods for each grid cell to eliminate the large influence of the annual cycle. To better compare the results of the trend analysis, the MERRA-2 and MISR datasets at high spatial resolution ($0.5\,°\times0.625\,°$ and $0.5\,°\times0.5\,°$, respectively) were re-gridded to the MODIS/Terra resolution of $1\,°\times1\,°$. Incomplete sampling from the satellite instruments may introduce biases in long-term trend analysis. Thus, to ensure the reliability of the trend analysis, each grid cell for the MISR and MODIS/Terra AODs was required to have valid data for at least 60% of the time period before regression was performed. Two-tailed Student's $t$-tests were used to assess the robustness of each trend estimate, and the criterion for statistical significance was set at the 95% confidence level.

Pearson's $R$ was used to measure the strength of the relationship between AOD, anthropogenic emissions, and meteorological parameters. MLR models of monthly MERRA-2 AODs were built for the 12 ROIs using emission factors, meteorological parameters, and both, as predictors. Four emission factors and 32 meteorological parameters were considered in the MLR models (Table 1). For each ROI, the MLR model could be expressed as

$$y = \beta_0 + \sum_{i=1}^{n}\beta_i x_i + \varepsilon, \qquad (4)$$

where $y$ is the standardized monthly AOD and $(x_1, \ldots, x_n)$ is the ensemble of standardized monthly explanatory variables. The standardized regression coefficient $\beta_i$ was determined by the least-squares method, and $\varepsilon$ is an error term.

In each step of the MLR model, a variable is considered to be moved or removed from the set of explanatory variables using the stepwise regression method to obtain the best model fit. In other words, for each step the model adds a significant ($P < 0.05$) explanatory variable to the model, it can be removed only if it is insignificant ($P > 0.1$) after adding or removing another variable. A similar model has been widely used to investigate the relationship between aerosols and meteorology (e.g., Yang et al., 2016; Tai et al., 2010).

Although the most important explanatory variables were obtained via the above stepwise MLR model, there might be multiple collinearities among different explanatory variables. In that situation, the standardized regression coefficient as an explanation of relative importance is unstable and misleading. To eliminate the influence of multi-collinearity, the variance inflation factor (VIF) (Altland et al., 2006)



was used to test whether there was a multi-collinearity problem among the variables.
VIF is often regarded as a measure of collinearity between each variable and another
variable in the model. VIF can be calculated from the following relationship:

$$\text{VIF} = \frac{1}{1 - R_i^2},\tag{5}$$

where $R_i^2$ is the coefficient of determination of linear regression between the $i$th
independent variable and other independent variables in the model. The present study
used a VIF threshold of 10, which is widely recommended in the literature (e.g., Hair
et al., 2010; Barnett et al., 2006; Field, 2005), to represent the maximum acceptability
of collinearity.
Finally, to better quantify the relative contributions of each independent
explanatory variable, which were obtained from the stepwise MLR model, to AOD
variability, the LMG method (Bi 2012; Grömping 2006; Lindeman et al., 2014) was
applied. This approach is one of the most advanced methods for determining the
relative importance of explanatory variables in a linear model and provides a
decomposition of the fraction of model-explained contributions (i.e., $R^2$) into
nonnegative contributions using semi-partial $R$ values. The LMG measure for the $i$th
regressor $x_i$ can be expressed as

$$\text{LMG}(x_i) = \frac{1}{p!} \sum_{r \ permutation} seqR^2(\{x_i\}|r),\tag{6}$$

where $r$ represents the $r$th permutation ($r = 1, 2,..., p!$), and $seqR^2(\{x_i\}|r)$ represents
the sequential sum of squares for the regressor $x_i$ in the ordering of the regressors in
the $r$th permutation.
For a detailed introduction to and description of the calculation process of the
LMG measure, refer to Grömping (2006). For all variables (including the AODs from
MERRA-2, MISR and MODIS/Terra, the meteorological variables from MERRA-2,
and the emission estimates from PKU), the regional mean was calculated by
averaging valid variable values over all grids within the twelve ROIs. For the seasonal
analysis, the four seasons were considered as follows: spring (March–April–May),
summer (June–July–August), autumn (September–October–November), and winter
(December–January–February).



## 3 Results and discussion

### 3.1 Assessing the performance of the MERRA-2 AOD datasets on the global scale

Although the official documentation points out that a large number of AOD observations have been assimilated into the system (Buchard et al., 2017), the global performance of MERRA-2 AOD is still unknown. Using all of the collected AERONET and CARSNET observations, the overall performance of the MERRA-2 AOD on a global scale was validated first. The results showed significant spatial agreement between MERRA-2 and ground-based AOD on the global scale, with an acceptable bias ($r = 0.84$, RMSE = 0.14, and MAE = 0.07) (Fig. 2). Moreover, Fig. 3 shows site-to-site comparisons of the three-hourly MERRA-2 AOD at 550 nm and the collocated AEROENT and CARSNET AOD observations, and a statistical summary of the comparison and the location information for each site are given in Tables S4 and S5. Globally, the MERRA-2 AOD datasets exhibited high $R$ values against ground-based observations: over 83.2%, 58.0% and 26.1% of sites had an $R$ greater than 0.6, 0.7 and 0.8, respectively; 80.4% and 47.1% of sites had an MAE greater than 0.1 and 0.05, respectively; and more than 65.3% and 85.1% of sites had an RMSE less than 0.1 and 0.2, respectively. These results indicated that, although MERRA-2 does not perform well in some individual regions, it does not affect the global accuracy of MERRA-2 as the latest global aerosol reanalysis dataset, especially in comparison with other satellite datasets. In addition, the obvious regional differences in the global performance of MERRA-2 AOD should not be overlooked. According to Fig. 3c1, the RMB was greater than 1 in the United States, southern South America and Australia, which indicates that MERRA-2 overestimates the AOD in these regions. In contrast, there clear underestimation was found in other regions, such as the Amazon Basin, Europe, SA, and SEA. This apparent underestimation (about 29%) in NC was further confirmed using additional ground-based AOD observations from CARSNET (reported in the following section). Notably, this underestimation seems to be systematic, as negative RMB was found in most parts of the Northern Hemisphere, except the United States. Such systematic underestimation over these regions is likely due to the lack of nitrate aerosols in the GOCART model (Buchard et al., 2017). Furthermore, the underestimation seems to be more prominent in high nitrate-emissions areas such as NC and SA.

To ensure the accuracy of inter-annual variations of AODs over different ROIs (as defined in Fig. 1), the regional performance of MERRA-2 AOD was evaluated by integrating all sites within each ROI (Figs. S2 and S3). Regionally, $R$ ranged from 0.7 to 0.95 among the 12 ROIs, with the highest $R$ (0.95) occurring in the ME and the lowest (0.7) in the EUS. Similar to the site-to-site RMB distribution, the RMB presented a systematic overestimation in the EUS of around 11%. In contrast, the RMB showed significant systematic underestimation in NC, SA, CF and SEA, with the degree of underestimation being 29%, 13%, 25% and 16%, respectively.



Significant differences in these regions were also supported by large MAEs of 0.25,
0.11, 0.08 and 0.12, respectively.
The MERRA-2 AOD datasets performed better over SA than over NC, which is
one of the most polluted areas in the world, in terms of a smaller MAE (0.11) and
RMSE (0.18) (Fig. S2f). The better performance over SA is likely due to more AOD
observations having been assimilated in MERRA-2 compared to over NC (Buchard et
al., 2017). For NEA, SC and WEU, MERRA-2 AOD generally compared well to
AERONET AOD, with the MAE being less than 0.1 and RMB greater than 0.93. For
the SD, results were relatively poor in that the MAE was greater than 0.1 and the
RMSE greater than 0.2. Besides, although MERRA-2 performed well in NWC when
only one AERONET site was used, after using additional CARSNET ground-based
observations it was found that the MERRA-2 AOD performance in NWC needs to be
improved (Fig. S3c). Notably, MERRA-2 was found to produce lower AOD than
AERONET, and the bias between them was more obvious for high AERONET AODs.
For instance, the MERRA-2 AODs over most polluted areas (such as the
anthropogenic aerosol–dominant regions of NC and SA and the biomass burning–
dominant regions of SEA and South America) were almost always lower than those of
AERONET when the AERONET AOD was greater than 1.5. This indicated that
MERRA-2 does not capture all high-AOD events well (such as serious haze events
over NC and SA, and frequent biomass burning events over SEA), due to the
following three reasons: (1) a relatively low quantity of ground-based-observed
aerosol data can be used for assimilation; (2) the MERRA-2 system model lacks an
adequate source of anthropogenic emissions with high temporal resolution; and (3) a
lack of nitrate aerosols in the GOCART model (Chin et al., 2002; Colarco et al., 2010;
Buchard et al., 2017).
To confirm whether MERRA-2 systematically underestimates the AOD over NC,
additional AOD observations from 12 CARSNET sites within NC were used for
comparison, and the corresponding statistical results for the site-by-site comparison
are given in Table S5. Compared with the results from using three AERONET sites as
a comparison, the results comparing CARSNET and MERRA-2 AOD showed a
similar pattern—that is, the underestimation of MERRA-2 AOD over NC is universal.
MERRA-2 underestimated the AOD at almost all CARSNET sites (Fig. 3c2 and Table
S5), with an overall MAE of 0.23, RMSE of 0.33, and underestimation of ~29% (Fig.
S3a). Similar results based on CARSNET observations in China have also been
reported in the literature (Song et al., 2018; Qin et al., 2018). Specifically, there was
higher agreement over SC compared with NC (Fig. S3b), mainly because nitrate
aerosols in China are mainly concentrated in industrially intensive areas such as
Henan, Shandong, Hebei, and the Sichuan Basin (Zhang et al., 2012). The lack of a
nitrate module in the GOCART model will cause further AOD uncertainty in these
above areas, which is the main reason behind the relatively low performance of
MERRA-2 AOD in these areas.
The purpose of this work was to study the inter-annual or inter-decadal variations
of AOD in different regions. Therefore, taking MODIS/Terra and MISR AOD as a
reference, the accuracy of MERRA-2 annual-mean AOD was evaluated at global and



regional scales (Figs. S4 and S5). Globally, the overall spatial correlations between
the MERRA-2 AOD and MODIS/Terra and MISR AOD datasets was found to be
quite acceptable, with no apparent disagreements in the annual AOD variations during
2001–2016 (Fig. S5). Besides, although an offset was found between MERRA-2,
MODIS/Terra and MISR in terms of absolute values of AOD in some ROIs, the
short-term tendency during the overlapping period was similar among the three
datasets (Fig. S4). Because the aerosol retrieval algorithm based on satellite
observation does not work well under cloudy conditions or for bright surfaces, there
are always numerous missing values in satellite-retrieved AOD datasets. In contrast,
not only is the accuracy of the MERRA-2 AOD dataset comparable with satellite
observations (Fig. S4), it also provides a complete AOD record from 1980 to the
present day. These reasons give confidence that the MERRA-2 aerosol dataset is
suitable for analysis of the variations in AOD. Thus, the AOD values from
MERRA-2's aerosol analysis fields, in combination with the AOD datasets derived
from two satellite sensors, were used to comprehensively analyze the spatiotemporal
variability of aerosols at global and regional scales.

## 3.2 Global AOD distribution and inter-annual evolution of regional AOD

Figure S6 shows the global annual- and seasonal-mean AOD distribution
calculated from the MERRA-2 AOD products during 1980–2016. Furthermore, the
distributional characteristics of the global annual-mean AOD from MERRA-2,
MODIS and MISR during the same period (2001–2016) are also compared in the
figure. The comparison shows that, although MISR underestimated the AOD (e.g., in
SA and eastern China), as expected because of insufficient sampling (Mehta et al.,
2016; Kahn et al., 2009), the three AOD products were generally closely consistent on
the global scale (also see Fig. S5). Generally, high AOD loading was mainly observed
in areas of high anthropogenic and industrial emissions, such as in eastern China and
India, and major source areas of natural mineral dust—particularly the Saharan,
Arabian and Taklimakan deserts.
Due to the seasonal variation of the atmospheric circulation driven by solar
radiation and the intensity of human activities in different regions, the global
distribution of AOD also shows obvious seasonal differences, with global aerosol
loading reaching its maximum in spring and summer. On the one hand, this can
mainly be attributed to the enhanced circulation in spring and summer, which
increases the likelihood of natural mineral dust from several major dust sources in the
Northern Hemisphere (i.e., the Sahara and Sahel, the Arabian Peninsula, Central Asia,
and the Taklimakan and Gobi deserts) being brought into the atmosphere; plus, along
the westerly belt, airflow dust can be transmitted to surrounding sea areas (such as the
strip of the northern tropical Atlantic stretching between West Africa and the
Caribbean, the Caribbean, the Arabian Sea, and the Bay of Bengal) and more remote
areas (such as South America, the Indo-Gangetic Plain, and the eastern coastal areas
of China, Korea, and Japan). On the other hand, higher temperatures and damp air in



summer can create favorable conditions for the hygroscopic growth and secondary
formation of aerosols (Minguillón et al., 2015; Zhao et al., 2018), which raises the
AOD in some areas, such as NC and northern India, dominated by anthropogenic
aerosol emissions in summer. Moreover, frequent local biomass-burning aerosol
emissions in central Africa during summer is the main cause of high AOD in the
region (Tummon et al., 2010).
In contrast, global aerosol loading is relatively low in autumn and winter. The
atmosphere in autumn and winter is generally more stable and vertical mixing is
weaker, and thus it is difficult for more aerosols—particularly natural mineral
dust—to be brought into the atmosphere, which leads to lower AOD in autumn and
winter. Nevertheless, the AOD in autumn in South America, SEA, SC and CF is
clearly high, which is mainly attributable to the emission of large amounts of fine
aerosol particles (i.e., BC and OC) from frequent biomass burning in these regions
(Thornhill et al., 2018; Ikemori et al., 2018; Chen et al., 2017). Notably, fine
particulate matter composed of sulfate–nitrate–ammonium aerosols, which is
produced by high-intensity anthropogenic activities in autumn and winter, is still the
main contributor to high AOD in eastern China and India (Gao et al., 2018; David et
al., 2018).
To better characterize the temporal evolution of regional AOD, the monthly mean
AODs over the 12 ROIs from 1980 to 2016 were calculated. As illustrated in Fig. 4,
the monthly regional AOD had large seasonal variability, in addition to varying
degrees of fluctuation in different periods. In areas dominated by smoke aerosols from
biomass burning (i.e., AMZ, CF and SEA), biomass-burning events tend to occur in
the warm season (May to October), leading to a more prominent monthly AOD at this
time of the year compared with the cold season (November to April). It is noteworthy
that MERRA-2 also captured several well-known forest-fire events, such as those in
Indonesia in 1983 and 1997, which have been proven to be mainly related to climatic
drying caused by El Niño and large-scale deforestation (Page et al., 2002; Goldammer
2007). In the CF region, the monthly mean maximum AOD experienced a
transformation process—that is, the monthly maximum AOD often occurred in June
and July before 2000, whereas after 2000 it occurred more frequently in August and
September. In the AMZ and SEA regions, September and October seems to be the two
most frequent months for the occurrence of high AOD values, but the magnitude of
AOD values has decreased in recent years.
In areas dominated by natural mineral dust aerosol (i.e., the SD, ME and NWC),
the monthly maximum AOD mainly occurred in March–August. Before 2000, there
were many anomalies of the AOD monthly maximum, which also implied frequent
sandstorms. In contrast, the frequency of monthly AOD anomalies decreased after
2000, which may be attributable to the reduced surface wind speed and increased
vegetation cover (Kim et al., 2017; Wang et al., 2018; An et al., 2018). Compared
with the areas dominated by smoke and dust aerosols, the seasonal differences of
AOD in the areas dominated by anthropogenic aerosol emissions appear to be smaller,
but their temporal evolution is more pronounced. In NEA, the monthly maximum
AOD often occurred in March–June, possibly related to the long-distance



transportation of sand and dust in the China–Mongolia deserts (Taklimakan and Gobi).
However, as the frequency of sandstorms has decreased in the past 10 years (An et al.,
2018), the monthly maximum AOD has also shown a downward trend. In NC and SA,
the monthly AOD has gradually expanded outward since 1980, indicating that AOD
has experienced a gradual increase. Monthly AOD had large seasonal variability in
the SC region, reaching its maximum in February–April. The increased aerosol
emissions from biomass burning in spring seem to be one of the main reasons for high
AOD in the SC region (Chen et al., 2017). For the EUS and WEU regions, the
characteristics of the monthly variation in AOD were similar—that is, large values of
AOD occurred in summer. With time, the monthly AOD showed a tendency to
gradually shrink inwards, suggesting AOD has experienced a significant decline over
the past few decades in the EUS and WEU. The main drivers of the inter-annual
variability of AOD over each ROI are discussed in detail in sections 3.5 and 3.6.
## 3.3  Global AOD trend maps
Annual and seasonal linear trends of the MERRA-2 AOD anomaly were
separately calculated for each $1° \times 1°$ grid cell for the whole of 1980–2016 period
(period 1) and for the first 18 years (1980–1997, period 2) and last 19 years (1998–
2016, period 3). Figure 5 shows the spatial distribution of these trends on the global
scale. Throughout period 1, the regions where annual AOD showed a significant
upward trend ($p < 0.05$) were mainly located in eastern China, SA, the ME, northern
South America, and the southern coastal areas of Africa, whereas some significant
downward trends were observed in the whole of Europe and the EUS. However,
compared with the annual trends, the seasonal AOD trends had obvious regional
differences in terms of their spatial distribution. For instance, a strong positive trend
throughout East Asia, including Korea and Japan, was found in spring. In summer,
there was a significant upward and downward AOD trend in north-central Russia and
the Amazon basin, respectively. In contrast, winter AOD had a significant downward
trend in the area north of 40 °N.
In the two different historical periods (i.e., period 2 and 3), these trends seem to
have experienced a remarkable shift. During period 2, the annual AOD had a
significant upward trend throughout the Southern Hemisphere, and similar upward
trends also existed in eastern and northwestern China. This upward trend in the
Southern Hemisphere, which was most likely associated with two giant volcano
eruption events in the early 1980s [El Chichón (Hofmann and Rosen 1983)] and early
1990s [Pinatubo volcanoes (Stenchikov et al., 1998; Bluth et al., 1992; Kirchner et al.,
1999)], is also reflected in the regional annual mean AOD time series shown in Fig.
S4. The eruptions led to a strong increase in volcanic ash and $SO_2$ emissions,
consequently increasing AODs from place to place via airflow transport, which was
captured accurately by MERRA-2. Meanwhile, AOD had a significant downward
trend throughout Europe and the EUS, which appears to be related to the reduction of
TSP and $SO_2$ emissions (see section 3.5). Seasonally, a significant upward trend
seems to be prevalent in all seasons in the Southern Hemisphere. Compared with
other seasons, the decline of AOD was more obvious in Europe and America. In



winter, except for the positive trend that still existed in the marine area of the Southern Hemisphere, the fluctuations in other regions were smaller and relatively stable.

During period 3, AOD began to show a significant upward trend in most regions, especially in SA, SEA, the ME, central Russia, the western United States, and northern South America, whilst still maintaining an upward trend in eastern China with greater intensity. These upward trends over SA, the ME and eastern China are in good agreement with the results of Hsu et al., (2012), who used SeaWiFS AOD records from 1997 to 2010. It is worth noting that the trends for the whole of Europe shifted from significantly positive to statistically insignificant, while the region that had shown a significant downward trend before 1997 in the EUS was also shrinking. Furthermore, the region showing a positive trend, prevailing in the Southern Hemisphere, shrunk dramatically. Similarly, the spatial distribution of the trend also had significant differences in different seasons of this period. In spring and winter, only significant upward trends could be observed on a global scale, mainly in eastern China, SA, the ME and South America. Conversely, significant downward trends were apparent in the EUS, Northwest Africa and central South America in summer. Additionally, it was also found that the region with a significant downward trend in Africa shifted from the northwest in summer to the southwest in autumn.

Ensuring the accuracy of AOD trends calculated by MERRA-2 is critical for quantifying the contribution of local emissions and meteorological factors to the inter-decadal variation of AOD in different regions. For comparison, the resulting annual and seasonal trends of the MERRA-2, MODIS/Terra, and MISR AOD anomaly over the whole globe were derived, using the same method, between 2001 and 2016; the results are shown in Fig. 6. This comparison shows that the AOD trends during 2001–2016 calculated by MERRA-2 in most regions of the world agreed well with the results of MODIS and MISR, on both annual and seasonal timescales. Although MERRA-2 assimilates MODIS and MISR at the same time, the relatively small difference between MERRA-2 and MISR may be mainly due to the insufficient sample size of MISR (MODIS produces three to four times more data than MISR) (De Meij et al., 2012).

For the annual trend, the significant upward trend observed by MODIS/Terra and MISR in SA and the ME and the significant downward trend observed in the EUS, WEU and central South America were consistent with the results of the MERRA-2 trend. Similar trends were reported in a previous study based upon 14 years (2001–2014) of observational records (Mehta et al., 2016). Similarly, upward trends also existed in spring, autumn and winter, while downward trends were also apparent in spring, summer and autumn. It should be noted that the trend signals calculated from MERRA-2 and MODIS/Terra were opposite in SC. The difference in sign associated with trends during 2001–2016 could mainly be due to the larger deviation between MERRA-2 and MODIS/Terra between 2001 and 2004 (Fig. S4c). The large deviation directly led to a reversal of trend throughout the period 2001–2016. This deviation may be related to the use of different versions of MODIS data: in the MERRA-2 AOD observing system, MERRA-2 assimilated the bias-corrected AOD derived from



MODIS radiances, Collection 5 (Buchard et al., 2017), and the MODIS data used in
this study was the latest collection (Collection 6.1). Different versions mean
differences in algorithms (Fan et al., 2017), which may affect the statistical error.

## 3.4 Regional AOD trends

To examine the spatial and temporal changes in more detail, the annual trend over
the globe and in the 12 ROIs, derived based upon MERRA-2 during periods 1, 2 and
3, were calculated. In addition, for comparison purposes, the regional trends in AODs
from MERRA-2, MODIS and MISR during 2001–2016 were also estimated. The
comparisons of the magnitudes of global annual trends with these regional trends are
summarized in Fig. 7 and Table S1. In general, the annual trends derived from
different datasets were small on the global scale. As indicated by the results in Fig. 7
and Table S1, the trend values were $-0.00068$ $yr^{-1}$ for the globe during period 1, with
statistical significance at the 95% confidence level. In contrast, no statistically
significant trend was detected at the global scale for period 2 (0.00050 $yr^{-1}$) or 3
(0.00038 $yr^{-1}$). Analyzing the global AOD trends during 2001–2016 from MERRA-2
and the two satellite datasets, it was found that the MERRA-2 trends were negligible,
whereas significant positive (negative) trends were found for MODIS (MISR).
However, the trends could be considerable on regional scales. For example, over
the anthropogenic aerosol–dominant regions for periods 1, 2 and 3, strong positive
trends were apparent over NEA, NC, SC and SA, while strong and statistically
significant negative trends were found over WEU and EUS. For biomass-burning
regions (SEA, CF and AMZ, but not CF, which had a negligible and insignificant
trend), there was a positive trend during periods 1, 2 and 3. For the mineral dust–
dominant regions, although there seemed to be an upward trend over the ME, the
estimated trends were not statistically significant for other areas, such as NWC and
the SD. During 2001–2016, the estimated MERRA-2 AOD trend in most ROIs (i.e.,
NEA, SA, ME, WEU, EUS, and AMZ) was comparable to and had the same sign as
the trend from both the MODIS and MISR sensors. However, it was opposite in sign
to the MISR data over NC, NWC and the SD, and to the MODIS data over SC, SEA
and CF during overlapping years.
In addition to the annual trend, the seasonal trend of AOD for different datasets in
different ROIs and different historical periods was also studied (Fig. S7 and Table S1).
Globally, negative trends were observed throughout the four seasons during period 1,
especially during summer, autumn and winter ($-0.00078$, $-0.00092$ and $-0.00097$ $yr^{-1}$,
respectively; statistically significant at the 95% confidence level). On the contrary,
there was a negative trend in period 2, although it was not significant. In the
subsequent period, period 3, the trend values shifted from negative to positive. The
positive trend was more significant in spring and autumn (0.00053 and 0.00070 $yr^{-1}$).
Regionally, strong positive trends were apparent over both NC and SC throughout the
four seasons during periods 1, 2 and 3. Strong upward trends were also found over SA.
These upward trends were most likely associated with an increase in urban/industrial
pollution in China and India. Meanwhile, some similar but relatively moderate
upward trends also existed over NEA in spring. In contrast, strong negative trends



were observed over the WEU and EUS regions, especially during spring, summer and
autumn. The negative trends over WEU and the EUS may partly have been due to a
decrease in polluting aerosols associated with emission control measures. A
statistically significant upward trend was also found over the SD, NWC and the ME
in spring during periods 1, 2 and 3 (0.00252, 0.00300 and 0.00463 yr$^{-1}$), respectively.
In contrast to the strong downward trends over AMZ in summer during periods 1, 2
and 3, there appeared to be upward trends in spring over AMZ and in winter over CF
and AMZ. When compared with the regional trends during 2001–2016 calculated by
the two satellite datasets, we found that the seasonal trends of MERRA-2 were highly
consistent with the satellite results in almost all regions, especially in spring and
autumn. It is worth noting that the trend differences among the three different datasets
in all four seasons still existed in NC and SC, and the differences had different
seasonal characteristics. For example, over NC, the most significant difference
occurred in spring and summer, whereas it occurred in summer and winter over SC.
Since the sign of a trend value often varies with the span of the calculation period,
it was necessary to evaluate the sliding trend of different periods to help examine the
time node of the changes more precisely. Therefore, sliding trend analyses were used
to present a more comprehensive analysis of annual trends over the 12 ROIs during
different historical periods (Fig. 8). These trends were calculated for all periods
starting each year from 1980 to 2007 and ending in 2016 with increments of at least
10 years. As shown in Fig. 8, in the EUS and WEU, the AOD experienced a large
decline up until the 1981–1990 period, and then the trend reversed moderately from
1984 to 1986, declined sharply from 1989 after a short increase from 1996 to 1999,
and then sustained a moderate downward trend in the last 17 years. A similar pattern
was found for NWC, SD and AMZ, although there was a stronger upward trend and
relatively weaker downward trend in the corresponding period. In SC and NC, the
AOD experienced a slight increase in the 1980s and a short-term decline around the
1990s, and then showed its largest positive trend since 1995 before reversing sharply
in the last 10 years. A similar evolution also existed in NEA and the ME, although the
intensities of the trends were relatively weak. In addition to the negligible downward
trend in the 1980s and 1990s, SA showed overall positive trends throughout the period.
Furthermore, in CF, a moderate increasing trend was detected from 1983 to 1985; then
in 1990, and the trends became relatively stable but unexpectedly showed sharp
increases after 1993, followed by a significant decline in the 2000s and reversal in the
last 10 years. The trends for SEA were much smaller and relatively stable. Also, note
that around 1985 and 1990 two distinct opposite trend signs were found in all regions.
These two unexpected trends indicated that large volcanic eruptions not only greatly
affect short-term changes in local aerosols, but also impose different degrees of
disturbance in long-term trends of aerosols in different regions of the world.
Furthermore, considering that aerosol concentration and composition usually
have strong seasonal cycles, the trends for each season were also calculated separately
and compared with the MODIS and MISR trends in the period of overlap (2001–
2016). Note that Fig. 9 only shows the evolution of seasonal and annual trends for
every 10-year period starting from 1980 to 2007 for MERRA-2, and from 2001 to



2007 for MODIS and MISR; refer to Figs. S8–11 for a fuller presentation of the
regional seasonal trend. For all regions, the trends for all seasons, except autumn in
SEA, CF and AMZ and spring in the SD, were in phase with the annual trend (also
see Fig. S12). In general, autumn trends over SEA, CF and AMZ were larger and
often out of phase, possibly attributable to the sudden increase in aerosol
concentration caused by biomass-burning events. Similarly, the spring trend over the
SD was also larger and more asynchronous than in other seasons. This phenomenon
can mainly be attributed to active spring dust events. In addition, compared with the
annual and seasonal regional trends during 2001–2016 (Fig. 7 and Fig. S7), the
decadal trends of MERRA-2 agreed better with the trend results from MODIS and
MISR. This implies that the trends can change relatively quickly with time.
Supporting evidence was also found from the strongest trends on both annual and
seasonal scales being mostly concentrated in the lower $y$-axis values (Fig. 9 and Figs.
S8–11). These results also highlight the importance of evaluating temporal shifts or
decadal AOD trends.

## 3.5 Response of inter-decadal variation in regional AOD to local emissions and meteorological parameters

Previous studies have shown that the inter-annual variations in regional AOD are
mainly controlled by changes in emissions and meteorological factors (De Meij et al.,
2012; Pozzer et al., 2015; Itahashi et al., 2012; Zhao et al., 2017; Chin et al., 2014).
First, the trends of the four emission factors (i.e., TSP, $SO_2$, BC, and OC) and their
correlations with AOD were calculated for the whole study period (1980–2014), as
well as for two individual periods (i.e., 1980–1997 and 1998–2014). Note that the
PKU global emissions inventories were only available for 1980–2014, which limited
our research to a relatively short period. Figures 10 and S13 show the linear trends in
emissions and their relationships with MERRA-2 AOD during 1980–2014,
respectively. The decreasing AOD trends over Europe and the EUS (see Fig. 5)
coincided with substantial reductions in the emissions of primary anthropogenic
aerosols (TSP and BC) and precursor gases ($SO_2$), corresponding to pollution controls
(Hammer et al., 2018; De Meij et al., 2012). This was also supported by significant
positive correlation between AOD and emissions in most regions of Europe and the
EUS (Fig. S13).
Positive trends in TSP and $SO_2$ were present over India and eastern China, which
explained the significant upward trend of AOD in these two regions. In addition,
eastern China and India experienced a shift in the emissions trend during the two
periods (Figs. S14 and 16). In 1980–1997, a significant upward trend existed in both
regions. In contrast, in 1998–2014, India at least maintained this upward trend for all
four emission factors, with it sometimes being even stronger, while the positive trends
in emissions of TSP and $SO_2$ over eastern China were interspersed with negative
trends. More importantly, the trend of BC and OC in eastern China reversed
completely. The shift in these emission trends in eastern China can mainly be
attributed to the implementation of multiple emission reduction policies (Zheng et al.,



2018). The reductions in emissions were at least partly responsible for the decreasing
trend of AOD in the NC and SC regions in the last 10 years (see Fig. 8). The trends in
primary BC emissions followed a similar pattern as the trends in OC emissions,
except there were positive trends over northeastern China and the positive (negative)
trends over CF, AMZ and SEA (WEU and SC) were lower in magnitude, reflecting
regional changes in fire activity. There were positive AOD trends in areas dominated
by biomass burning (especially in CF and SEA), in response to increased BC and OC
emissions. Because human activities are scarce in desert areas, there was no direct
relationship between AOD and emissions, as expected. Therefore, this highlights the
importance of studying how natural factors (here, this refers to meteorological
parameters) control the inter-annual variation of AOD in different desert areas.
Furthermore, it is worth noting that in the two short periods (especially 1998–2014),
these regions with significant positive correlation shrunk and were no longer
significant (Figs. S15 and 17), suggesting other factors such as meteorological
parameters might be driving the inter-annual trend of regional AOD.

To investigate the roles of meteorological parameters in the decadal variation of
AOD, Pearson's $R$ values between AOD and meteorological parameters (a total of 32;
see Table 1) and over the 12 ROIs for the three periods (i.e., 1980–2014, 1980–1997
and 1998–2014) were calculated. Some of these meteorological variables, such as
surface precipitation, surface wind speed, wind velocity, RH, and surface wetness,
have been shown before to be correlated with regional AOD (Klingmüller et al., 2016;
Pozzer et al., 2015; Chin et al., 2014; He et al., 2016). Correlation analysis showed
similar correlation patterns between AOD and meteorological parameters for the three
different periods over all ROIs. During the period 1998–2014, the correlation was
generally stronger than in the other two periods (see Fig. S18), suggesting
meteorological factors may have played a more important role in this period. In
addition, these correlations seemed to be similar in regions dominated by the same
aerosol type. For example, in the mineral dust–dominated regions (i.e., NWC, ME and
the SD), AOD had a significant positive (negative) correlation with near-surface wind
speed (soil moisture), suggesting that surface wind speed and soil moisture may be the
main factors controlling the dust cycle, which is consistent with previous studies in
the ME (Klingmüller et al., 2016). In the biomass burning–dominated regions (i.e.,
SEA, CF and AMZ), AOD had a significant negative correlation with
humidity-related meteorological parameters (such as surface precipitation, RH, and
soil moisture), implying that ambient humidity (including the atmosphere and soil)
may be a direct correlation factor in controlling the frequency of biomass-burning
events. In contrast, in the regions dominated by anthropogenic aerosols, the
correlation was regionally dependent, and their signs differed from place to place.

Correlation analysis cannot directly identify the main factors affecting the
inter-decadal change of AOD in different regions. Here, MLR models were used to
diagnose the influences of local anthropogenic emissions and other meteorological
parameters on the inter-decadal variation of AOD over the 12 ROIs. Figure 11 shows
the time series of monthly mean MERRA-2 and MLR model–predicted normalized
AOD anomalies, which used the emission factors, meteorological parameters, and





both, as input predictors, respectively, over the 12 ROIs for the whole study period
(1980–2014). Similar comparisons for the two individual periods (i.e., 1980–1997 and
1998–2014) are also presented in Figs. 19 and 20, respectively. Table S2 summarizes
the predictors included in the MLR models and their performance for the three
different periods over each ROI. The MLR models with both emissions and
meteorological parameters as predictors generally reproduced the AOD values in most
regions during 1980–2014, except for high AOD values (Fig.11), which is discussed
below. For all the ROIs, the MLR models explained most of the MERRA-2 AOD
variability ($R^2$ = 0.42–0.76). However, when meteorology and emissions alone were
used as predictors, there were considerable differences in different ROIs. When
emission factors alone were used as the predictor, it could account for more than 35%
of the AOD variability in regions dominated by anthropogenic aerosols and biomass
burning [except NEA (14%)], with the largest explanation occurring in NC (58%). In
contrast, in the mineral dust–dominated regions (the SD and ME), emission factors
contributed little (< 0.05%) to the inter-annual variation in AOD (Figs. 11g and i).
Moreover, emission factors contributed 37% of the AOD variability in NWC, which is
mainly because of the strong anthropogenic emission sources in northern Xinjiang
(mainly encompassing Urumqi, Korla, Kashgar, etc.). However, compared with
meteorological factors, emissions were not the main factors driving the inter-annual
change of AOD (Fig. 11e).
On the other hand, when meteorological factors were used as predictors in the
MLR models, it was surprising that they explained a larger proportion of the AOD
changes in all ROIs, except NC and SEA, where emission factors accounted for
slightly lower AOD changes of 42% and 33%, respectively. Further analysis indicated
that this difference in contribution between emissions and meteorology seemed to be
greater for the two shorter periods of 1980–1997 and 1998–2017 (see Figs. S19 and
20). Besides, it should also be noted that the total explained variances of the MLR
model for 1980–1997 were generally lower than those of the MLR model for 1998–
2014, in all ROIs. The difference can be explained by two reasons: (1) a greater
number of high AOD anomaly values occurred during the period 1980–1997 (Figs. 11
and S19), especially in relation to the two volcanic eruption events in the 1980s and
1990s, which directly reduced the total explained variances of the MLR model,
because the model only considers the inter-decadal variations of local emissions and
meteorological factors, and the large-scale transport of pollutants is not considered;
and (2) meteorology and emissions were confirmed to explain more AOD changes
during the period 1998–2014.

## 799 3.6 Relative contributions of local emissions and meteorological

## 800 parameters to inter-decadal variations of regional AOD

Application of the LMG method (see Data and Methods section) to the MLR
model allowed the relative contributions of each anthropogenic emission type and
meteorological factor to the inter-decadal variations or trend of regional AOD to be
quantified. Figure 12 shows the relative contributions of the local emissions and





meteorological factors to the changes in regional AOD for the period 1980–2014, as well as for 1980–1997 and 1998–2014, using both emissions and meteorology as predictors in the MLR model. During the period 1980–2014, over the anthropogenic aerosol–dominant regions, $SO_2$ was the dominant emissions driving factor, explaining 24.9%, 15.2%, 32.6%, 21.7% and 12.7% of the variance of AOD over NC, SC, SA, WEU and the EUS, respectively (also see Table S3). The above results also confirm that particulate sulfate is the main contributor to fine-mode AOD in anthropogenic aerosol–dominant regions (Itahashi et al., 2012; David et al., 2018). Meanwhile, wind speed (including surface and upper wind speed) was the dominant meteorological driving factor, explaining 11.4%, 14.2 % and 17.9% of the variance of AOD over NC, SC and the EUS, respectively. In addition, planetary boundary layer height, temperature (including surface temperature, upper temperature, and the temperature difference between the surface and upper atmosphere) and RH (including surface and upper RH) were the strongest meteorological driving factors over NEA, SA and WEU, contributing 30.2%, 15.9% and 21.5%, respectively.

On the contrary, over the biomass burning–dominant regions, BC (OC) was the dominant emissions driving factor over SEA (AMZ), explaining 27.7% (24.0%) of the variance of AOD. Meanwhile, soil moisture and RH were the top meteorological driving factors over SEA and AMZ, and CF, contributing 11.7% and 35.5%, and 28.5%, respectively. Furthermore, over the dust-dominant regions, WS was the strongest meteorological driving factor, explaining 30.3% and 29.8% of the variance in AOD over NWC and the SD, respectively. Different from WS being the primary meteorological driving factor over NWC and the SD, it was the second most important factor over the ME, while sea level pressure was the primary driving factor, accounting for 60.9% of the variation in AOD. This large variance explained by sea level pressure and significant anti-correlations of the AOD with it (see Fig. S18c), further confirms the previous studies' findings that frequent sandstorms over the ME often correspond to large horizontal pressure gradient differences caused by the enhanced high-pressure system across the eastern Mediterranean Sea and enhanced low-pressure system across Iran and Afghanistan (Hamidi et al., 2013; Yu et al., 2016).

By comparing the estimated results of the two independent study periods (i.e., 1980–1997 and 1998–2014), it was found that in almost all ROIs (except NC and AMZ), meteorological factors contributed a larger explained proportion of AOD changes during 1998–2014, which indicates that meteorological factors seem to be becoming increasingly more important in dominating the inter-decadal change of regional AOD. It is worth noting that, in addition to the increased explained proportion of $SO_2$ and BC, among these meteorological factors, the role of diffusion-related parameters (such as horizontal and vertical WS, representing horizontal and vertical diffusion, respectively) seems to be the most prominent. This is consistent with the findings of Gui et al. (2019), who found WS to be the dominant meteorological driver for decadal changes in fine particulate matter over SC, based upon a 19-yr record of satellite-retrieved fine particulate matter data (1998–2016).





## 4 Conclusions and implications


This paper presents a comprehensive assessment of the global and regional AOD
trends over the past 37 years (1980–2016), based on the reanalysis MERRA-2 AOD
dataset. AOD observations from both AERONET and CARSNET stations were used
to assess the performance of the MERRA-2 AOD dataset on global and regional
scales prior to calculating the global and regional AOD trends. Satellite retrievals
from MODIS/Terra and MISR were then used to estimate the AOD annual and
seasonal trends and compare them with the MERRA-2 results. Finally, the stepwise
MLR and LMG methods were jointly applied to quantify the influences of emission
factors and meteorological parameters on the inter-decadal changes in AOD over 12
ROIs during the three periods of 1980–2014, 1980–1997 and 1998–2014.
Results showed that the MERRA-2 AOD was comparable in accuracy with the
satellite-retrieved AOD, albeit there was slight underestimation on the global scale
when compared with the in-situ AERONET and CARSNET AOD. MERRA-2 was
proven to be capable of estimating the long-term variability and trend of AOD, owing
to its good accuracy and continuous and complete spatiotemporal resolution. It was
revealed that, in general, MERRA-2 was able to quantitatively reproduce the AOD
annual and seasonal trends (especially decadal trends) during the overlapping years
(2001–2016), as observed by the MODIS/Terra, albeit some discrepancies (caused by
the insufficient sample size) were found when compared to MISR. The resulting trend
analyses based upon the MERRA-2 data from 1980 to 2016 showed that the global
annual trend of AOD during this period, although significantly ($p < 0.05$) weakly
negative (i.e., $-0.00068$ yr$^{-1}$), was essentially negligible when compared to the
magnitudes of regional AOD trends. On regional scales, sliding trend analyses
suggested that the inter-decadal trends of AOD in different periods could be
significantly different. It was noted that, during the entire study period (1980–2016),
the EUS and WEU showed a non-monotonous decreasing trend accompanied by
occasional fluctuations in the 1980s and 1990s, responding to the decrease in
pollutant emissions, but the intensity of this downward tendency has slowed over the
recent decade. In contrast, AODs in NC and SC experienced a sustained and
significant upward trend before ~2006, and then the trend shifted from upward to
downward due to the Chinese government's emissions-reduction policy. In addition to
the negligible downward trend in the 1980s and 1990s, SA showed overall significant
positive trends throughout the study period. Moreover, the two large volcanic
eruptions that occurred in the 1980s and 1990s not only greatly affected the
short-term changes in local aerosol loading, but also impacted significantly on the
inter-annual trend of the regional AOD around the world. This highlights the
importance of examining the effects of trans-regional pollutant transport on decadal or
temporal shifts in local AOD trends.
To diagnose the influences of local anthropogenic emissions and other
meteorological parameters on the inter-decadal variation of regional AODs, statistical
MLR models that estimated AOD monthly values over each ROI as a function of local
emissions factors and various meteorological variables were developed. The modeled



AODs using emission factors, meteorological parameters, and both, as input predictors in the MLR models were compared during three individual periods (i.e., 1980–2014, 1980–1997 and 1998–2014). In general, the MLR models with both emissions and meteorological parameters as predictors could account for 42%–76% of the variability of the MERRA-2 AOD, depending on the ROI. However, when meteorology and emissions alone were used as predictors, there were considerable differences in different ROIs. During 1980–2014, compared with the emission factors (0%–56%), it was found that meteorological parameters explained a larger proportion of the AOD changes (20.4%–72.8%) over all ROIs (except NC and SEA). Besides, further analysis also showed that this dominant driving role of meteorological parameters was stronger during the other two periods.

The LMG method for MLR models suggested that $SO_2$ was the dominant emissions driving factor, explaining 24.9%, 15.2%, 32.6%, 21.7% and 12.7% of the variance of AOD over NC, SC, SA, WEU and the EUS, respectively. In contrast, BC (OC) was the dominant emissions driving factor over SEA (AMZ), explaining 27.7% (24.0%) of the variance of AOD. For meteorological driving factors, over the mineral dust–dominant regions, WS was the top driving factor, explaining 30.3% and 29.8% of the variance of AOD over NWC and the SD. Meanwhile, soil moisture and RH were the strongest meteorological driving factors over SEA and AMZ, and CF, contributing 11.7% and 35.5%, and 28.5%, respectively. Notably, the performance of the MLR model in 1980–1997 was significantly worse than that in 1998–2014, which can mainly be attributed to the fact that the statistical model used in this study did not take into account the impact of trans-regional transport. Consequently, the model failed to capture the abnormally high values of regional AOD caused by trans-regional transport during 1980–1997. Finally, deeper insight into the influence of emissions and meteorological factors, as well as the influence of atmospheric transport, on the inter-decadal change in regional AOD, will be provided in future modeling studies.

## Data availability:

The CARSNET AOD dataset used in the study can be requested by contacting the corresponding author.

## Competing interests:

The authors declare that they have no conflict of interest.

## Author contribution:

All authors contributed to shaping up the ideas and reviewing the paper. HC, KG and XZ designed and implemented the research, as well as prepared the manuscript; HC, KG and YW contributed to analysis of the MERRA-2, MODIS and MISR dataset; HC, XX, BNH, PG, and EGA contributed to the CARSNET data retrieval; HC, KG, YW, HW, YZ, and HZ carried out the CARSNET observations; XX, BNH, PG, and EGA





provided constructive comments on this research

## Acknowledgements:

This work was supported by grants from the National Science Fund for Distinguished
Young Scholars (41825011), the National Key R & D Program Pilot Projects of China
(2016YFA0601901 and 2016YFC0203304), National Natural Science Foundation of
China (41590874), the CAMS Basis Research Project (2017Z011), the European
Union Seventh Framework Programme (FP7/2007-2013) under grant agreement no.
262254, and the AERONET-Europe ACTRIS-2 program, European Union's Horizon
2020 research and innovation programme under grant agreement no. 654109. NASA's
global modeling and assimilation office is gratefully acknowledged for making the
MERRA-2         aerosol         reanalysis         publicly         accessible
(https://disc.gsfc.nasa.gov/daac-bin/FTPSubset2.pl). Thanks are also extended to the
PKU emissions inventory research group (http://inventory.pku.edu.cn/home.html) and
AERONET networks (https://aeronet.gsfc.nasa.gov/) for making their data available
online, as well as the GES-DISC for providing gridded AOD products of MODIS and
MISR through their Giovanni website (https://giovanni.gsfc.nasa.gov/giovanni/).

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



**Table captions:**

**Table 1.** Prediction variables used in the stepwise MLR models.

**Figure captions:**

**Figure 1.** Geographical locations of the AERONET (yellow dots) and CARSNET sites (magenta dots) used in this work. The red boxes represent the 12 regions of interest selected in this study: Northeast Asia (NEA), northern China (NC), southern China (SC), Southeast Asia (SEA), Northwest China (NWC), South Asia (SA), Middle East (ME), western Europe (WEU), Sahara Desert (SD), Central Africa (CF), eastern United States (EUS), and Amazon Zone (AMZ).

**Figure 2.** Validation of the combined AERONET and CARSNET AODs against the three-hourly MERRA-2 AOD on the global scale. The color-coded dots indicate the number of samples. The solid red line is the line of best fit and the black dashed line is the 1:1 line.

**Figure 3.** Comparison of the three-hourly MERRA-2 AOD datasets with AOD observations of 468 AERONET sites worldwide and 37 CARSNET sites in China: site performance maps for the (a1) correlation coefficient ($R$), (b1) mean absolute error (MAE) and root-mean-square error (RMSE), and (c1) relative mean bias (RMB) between MERRA-2 AOD and ground-based AERONET AOD observations. Panels (a2) to (c2) are enlarged site performance maps for $R$, MAE and RMSE, and RMB, respectively, using the CARSNET observations as reference. The size of the circles in (b1) and (b2) represent the RMSE and their inner color represents the MAE. Panels (a3), (b3), (b4) and (c3) are frequency distribution histograms for the $R$, RMSE, MAE and RMB between MERRA-2 and all ground-based observations incorporating AERONET and CARSNET, respectively. Note that all sites within each region of interest (ROI) are integrated to assess the accuracy of the MERRA-2 AOD dataset in that area. The performance of the MERRA-2 AOD dataset in each ROI is illustrated in Figs. S2 and S3.

**Figure 4.** Temporal evolution of regional monthly averaged AOD for the 12 regions of interest. Each year is represented by an irregular ring with 12 directions. Each direction of the ring represents a specific month; the distance from the center of the ring represents the regional monthly mean AOD value; and the color of the ring represents the year. A special ring colored cyan represents the monthly mean AOD for the period 1980–2016.

**Figure 5.** Spatial distributions of the linear trends in annual and seasonal MERRA-2 AOD calculated from the time series value of the de-seasonalized monthly anomaly during (a) 1980–2016, (b) 1980–1997, and (c) 1998–2016. Only trend values with statistical significance at the 95% confidence level are shown.

**Figure 6.** Spatial distributions of annual and seasonal trends in AOD calculated from the time series value of the de-seasonalized monthly anomaly from (a) MERRA-2, (b) MODIS/Terra, and (c) MISR between 2001 and 2016. Only trend values with statistical significance at the 95% confidence level are shown.





**Figure 7.** Inter-comparisons of global and regional annual trends in AOD calculated from the time series value of the de-seasonalized monthly anomaly of MERRA-2, MODIS/Terra and MISR, during the four periods of 1980–2016, 1980–1997, 1998–2016, and 2001–2016. Error bars represent the uncertainty associated with the calculated trend. The trend bars with shadow indicate statistical significance at the 95% confidence level.

**Figure 8.** Sliding-window trend analyses of the annual mean MERRA-2 AOD from 1980 to 2016 over the 12 ROIs (see Fig. 1 for names and locations of regions), with at least 10 years used to calculate trends. The *x*-axis and *y*-axis indicate the start year and the length of the time series to calculate the trend, respectively. The colors of rectangles represent the intensity of the trend (units: /year), and those with black 'x' signs indicate linear trends above the 95% significance level.

**Figure 9.** Temporal evolution of sliding decadal trends in the annual and seasonal mean AOD from MERRA-2, MODIS/Terra and MISR over the 12 ROIs. The trends were calculated for each 10-year interval from 1980 to 2007 for MERRA-2, and from 2001 to 2007 for MODIS/Terra and MISR. The colors of the rectangles represent the intensity of the decadal trend (units: /year), and those with black 'x' signs indicate linear trends above the 95% significance level.

**Figure 10.** Spatial distributions of linear trends (units: $kg/km^2/year$) in total anthropogenic emissions of total suspended particles (TSP), $SO_2$, black carbon (BC), and organic carbon (OC) during 1980–2014 derived from the Peking University emissions inventory (http://inventory.pku.edu.cn/) (Huang et al., 2014). Only linear trend values with statistical significance at the 95% confidence level are shown.

**Figure 11.** Time series of MERRA-2 (in black) and modeled AOD monthly normalized anomalies from 1980 to 2014 over the 12 regions of interest. The coefficient of determination ($R^2$) of the regression fit of the stepwise MLR model with emission factors (in blue), meteorology (in green), and both emissions and meteorology (in red) as predictors are given in the top-right of each panel.

**Figure 12.** The LMG method–estimated relative contributions (%) of total variances in the stepwise MLR model explained by the local emission factors (left-hand bars) and meteorological variables (right-hand bars) over the 12 regions of interest during three periods: (a) 1980–1997 (top panel); (b) 1998–2014 (middle panel); and (c) 1980–2014 (bottom panel). Note that meteorological parameters were combined as follows: temperature, T ($T_s$, $T_{850}$, $T_{700}$, $T_{500}$, $dT_{900-s}$, $dT_{850-s}$); geopotential height, GH ($GH_{850}$, $GH_{700}$, $GH_{500}$); relative humidity, RH ($RH_s$, $RH_{850}$, $RH_{700}$, $RH_{500}$); vertical velocity, Ome ($Ome_{850}$, $Ome_{700}$, $Ome_{500}$); and wind speed, WS ($U_{850}$, $U_{700}$, $U_{500}$, $V_{850}$, $V_{700}$, $V_{500}$, $WS_s$, $WS_{850}$, $WS_{700}$, $WS_{500}$, $VWS_{500-850}$). Refer to Table S3 for the detailed relative contributions of each variable in the stepwise MLR models.



Table 1. Prediction variables used in the stepwise MLR models.

| Data type | Variables | Predictors used in the stepwise MLR model[a] | Data source |
|---|---|---|---|
| Emission factors | TSP | Gridded monthly total emissions of total suspended particles | Peking University global emissions inventories at $1° \times 1°$ horizontal resolution (http://inventory.pku.edu.cn/home.html) |
| | $SO_2$ | Gridded monthly total emissions of sulfur dioxide | |
| | BC | Gridded monthly total emissions of black carbon | |
| | OC | Gridded monthly total emissions of organic carbon | |
| Meteorological parameters | Pre | Gridded monthly total surface precipitation | MERRA-2 reanalysis dataset at $0.5° \times 0.625°$ horizontal resolution (https://disc.gsfc.nasa.gov/daac-bin/FTPSubset2.pl) |
| | PBLH | Gridded monthly mean planetary boundary layer height | |
| | SM | Gridded monthly mean soil moisture at surface | |
| | SLP | Gridded monthly mean sea level pressure | |
| | CLF | Gridded monthly mean cloud fraction | |
| | $T_s$ | Gridded monthly mean surface temperature | |
| | T | Gridded monthly mean 850-, 700- and 500-hPa temperature | |
| | dT | Gridded monthly mean temperature difference between 900 hPa and the surface, and 850 hPa and the surface | |
| | GH | Gridded monthly mean 850-, 700- and 500-hPa geopotential height | |
| | $RH_s$ | Gridded monthly mean surface relative humidity | |
| | RH | Gridded monthly mean 850-, 700- and 500-hPa relative humidity | |
| | Ome | Gridded monthly mean 850-, 700- 500-hPa vertical velocity | |
| | U | Gridded monthly mean 850-, 700- and 500-hPa zonal wind | |
| | V | Gridded monthly mean 850-, 700- and 500-hPa meridional wind | |
| | $WS_s$ | Gridded monthly mean surface wind speed | |
| | WS | Gridded monthly mean 850-, 700- and 500-hPa wind speed | |
| | $VS_{500-850}$[b] | Gridded monthly mean vertical wind shear between 500 and 850 hPa | |

[a]Units: $g/km^2$ (TSP, $SO_2$, BC, OC); $kg/m^2/s$ (Pre); m (PBLH, GH); 1 (SM, CLF); Pa (SLP); K (T, dT); % (RH); pa/s (Ome); and m/s (U, V, WS, $VWS_{500-850}$)
[b] $VWS_{500-850}$ was calculated as $\sqrt{(U_{500} - U_{850})^2 + (V_{500} - V_{850})^2}$





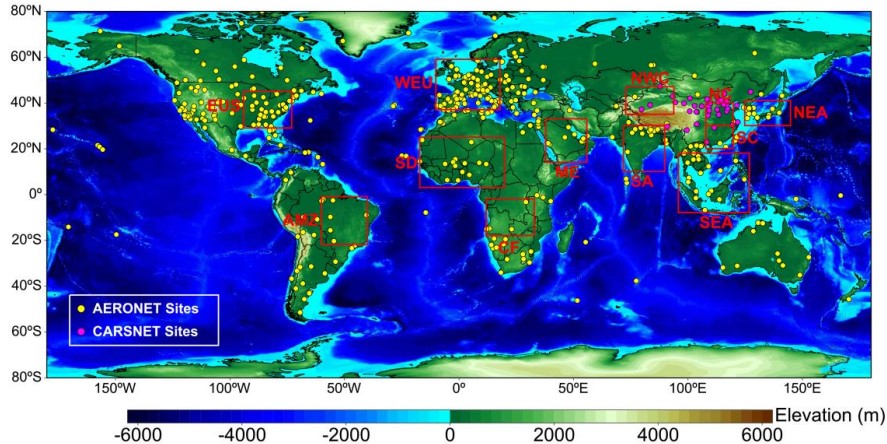


**Figure 1**. Geographical locations of the AERONET (yellow dots) and CARSNET sites (magenta dots) used in this
work. The red boxes represent the 12 regions of interest selected in this study: Northeast Asia (NEA), northern
China (NC), southern China (SC), Southeast Asia (SEA), Northwest China (NWC), South Asia (SA), Middle East
(ME), western Europe (WEU), Sahara Desert (SD), Central Africa (CF), eastern United States (EUS), and Amazon
Zone (AMZ).




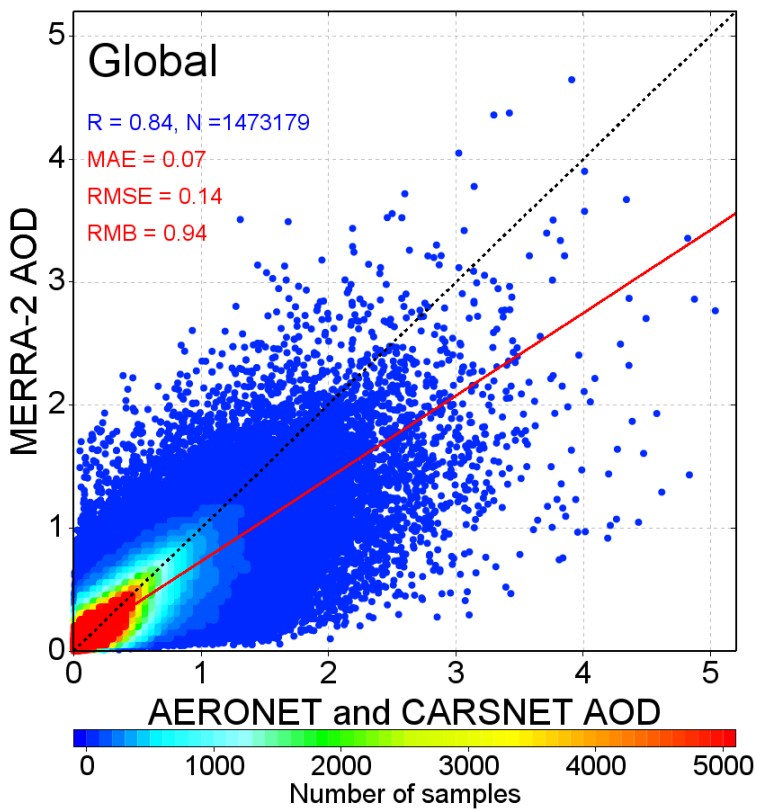


**Figure 2.** Validation of the combined AERONET and CARSNET AODs against the three-hourly MERRA-2 AOD
on the global scale. The color-coded dots indicate the number of samples. The solid red line is the line of best fit
and the black dashed line is the 1:1 line.




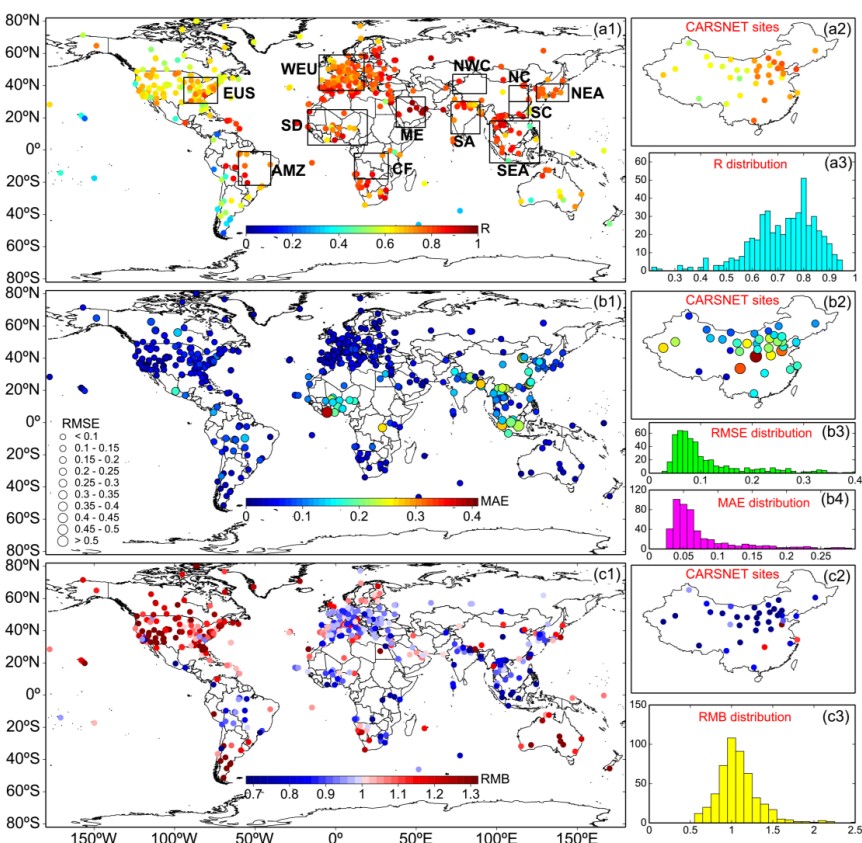


**Figure 3**. Comparison of the three-hourly MERRA-2 AOD datasets with AOD observations of 468 AERONET
sites worldwide and 37 CARSNET sites in China: site performance maps for the (a1) correlation coefficient ($R$),
(b1) mean absolute error (MAE) and root-mean-square error (RMSE), and (c1) relative mean bias (RMB) between
MERRA-2 AOD and ground-based AERONET AOD observations. Panels (a2) to (c2) are enlarged site
performance maps for $R$, MAE and RMSE, and RMB, respectively, using the CARSNET observations as reference.
The size of the circles in (b1) and (b2) represent the RMSE and their inner color represents the MAE. Panels (a3),
(b3), (b4) and (c3) are frequency distribution histograms for the $R$, RMSE, MAE and RMB between MERRA-2
and all ground-based observations incorporating AERONET and CARSNET, respectively. Note that all sites within
each region of interest (ROI) are integrated to assess the accuracy of the MERRA-2 AOD dataset in that area. The
performance of the MERRA-2 AOD dataset in each ROI is illustrated in Figs. S2 and S3.





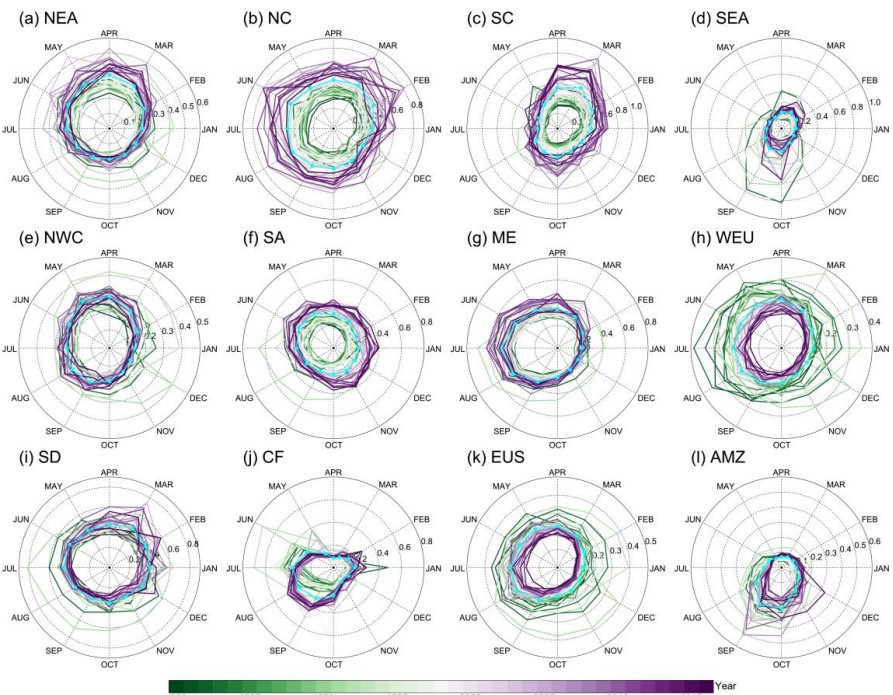

**Figure 4.** Temporal evolution of regional monthly averaged AOD for the 12 regions of interest. Each year is represented by an irregular ring with 12 directions. Each direction of the ring represents a specific month; the distance from the center of the ring represents the regional monthly mean AOD value; and the color of the ring represents the year. A special ring colored cyan represents the monthly mean AOD for the period 1980–2016.





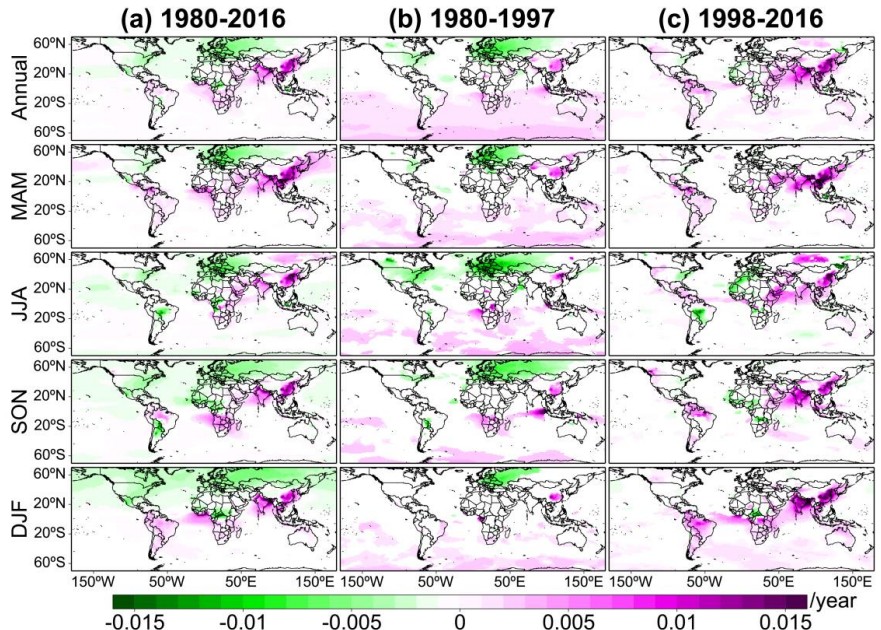

**Figure 5.** Spatial distributions of the linear trends in annual and seasonal MERRA-2 AOD calculated from the time series value of the de-seasonalized monthly anomaly during (a) 1980–2016, (b) 1980–1997, and (c) 1998–2016. Only trend values with statistical significance at the 95% confidence level are shown.





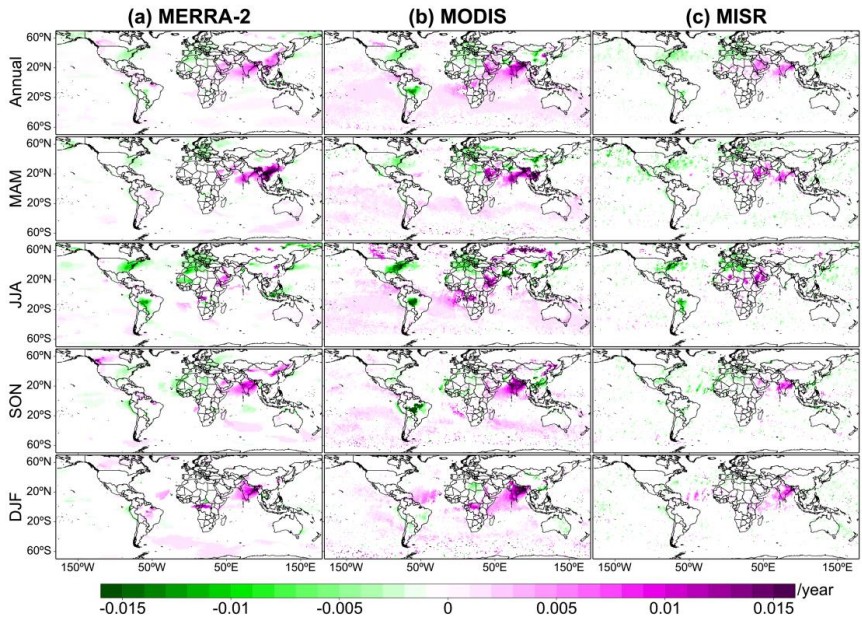

**Figure 6.** Spatial distributions of annual and seasonal trends in AOD calculated from the time series value of the de-seasonalized monthly anomaly from (a) MERRA-2, (b) MODIS/Terra, and (c) MISR between 2001 and 2016. Only trend values with statistical significance at the 95% confidence level are shown.



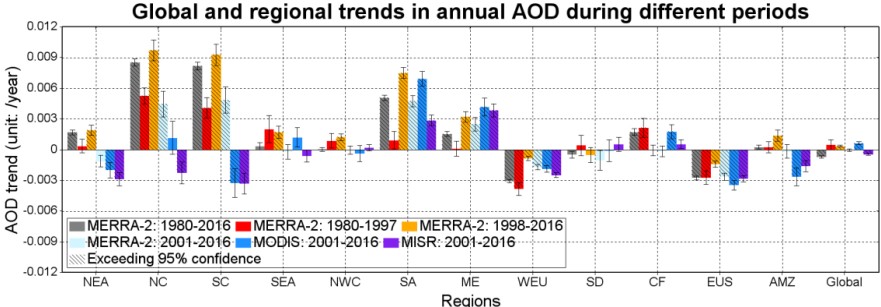

**Figure 7.** Inter-comparisons of global and regional annual trends in AOD calculated from the time series value of the de-seasonalized monthly anomaly of MERRA-2, MODIS/Terra and MISR, during the four periods of 1980–2016, 1980–1997, 1998–2016, and 2001–2016. Error bars represent the uncertainty associated with the calculated trend. The trend bars with shadow indicate statistical significance at the 95% confidence level.

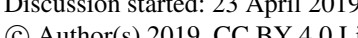
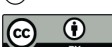


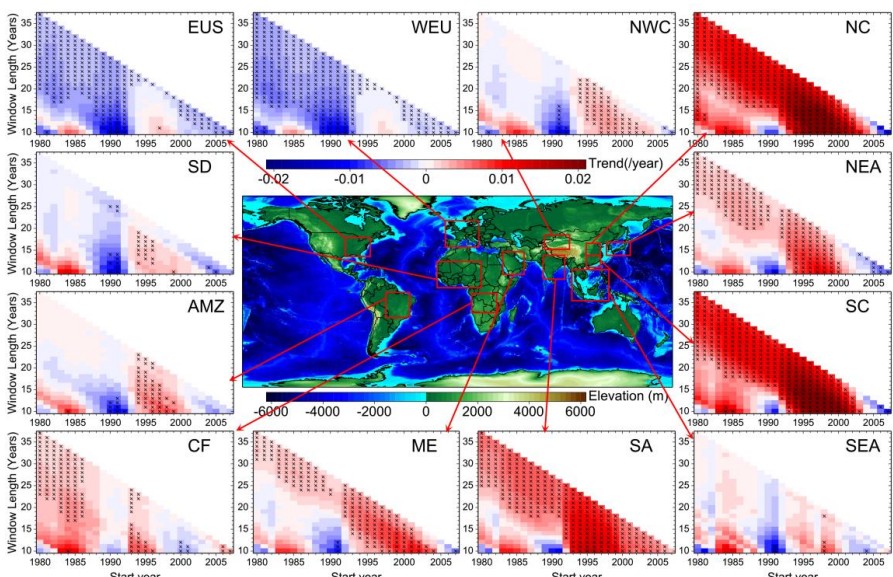


**Figure 8.** Sliding-window trend analyses of the annual mean MERRA-2 AOD from 1980 to 2016 over the 12 ROIs
(see Fig. 1 for names and locations of regions), with at least 10 years used to calculate trends. The *x*-axis and *y*-axis
indicate the start year and the length of the time series to calculate the trend, respectively. The colors of rectangles
represent the intensity of the trend (units: /year), and those with black 'x' signs indicate linear trends above the 95%
significance level.


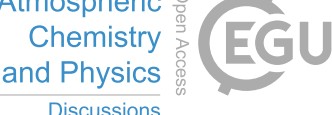

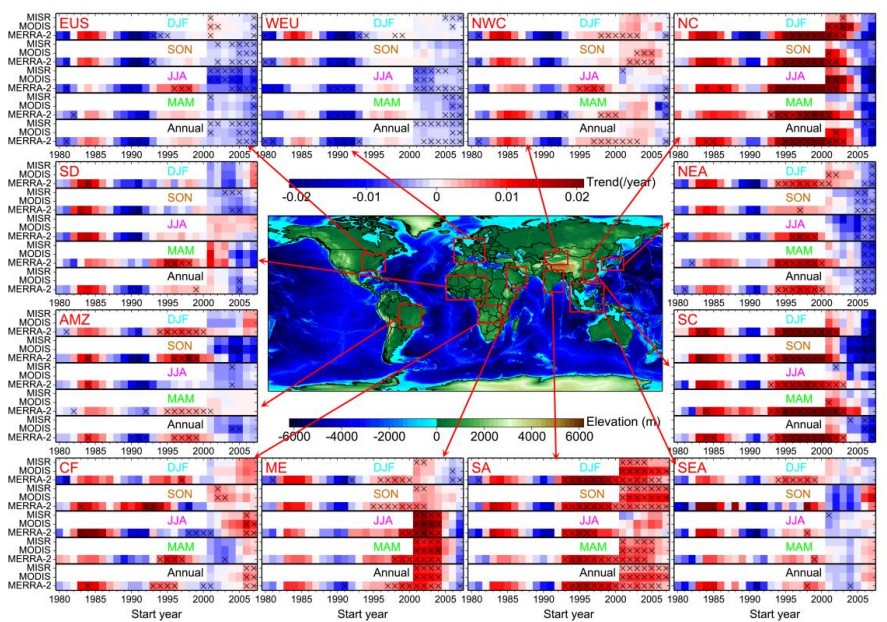

**Figure 9.** Temporal evolution of sliding decadal trends in the annual and seasonal mean AOD from MERRA-2,
MODIS/Terra and MISR over the 12 ROIs. The trends were calculated for each 10-year interval from 1980 to 2007
for MERRA-2, and from 2001 to 2007 for MODIS/Terra and MISR. The colors of the rectangles represent the
intensity of the decadal trend (units: /year), and those with black 'x' signs indicate linear trends above the 95%
significance level.





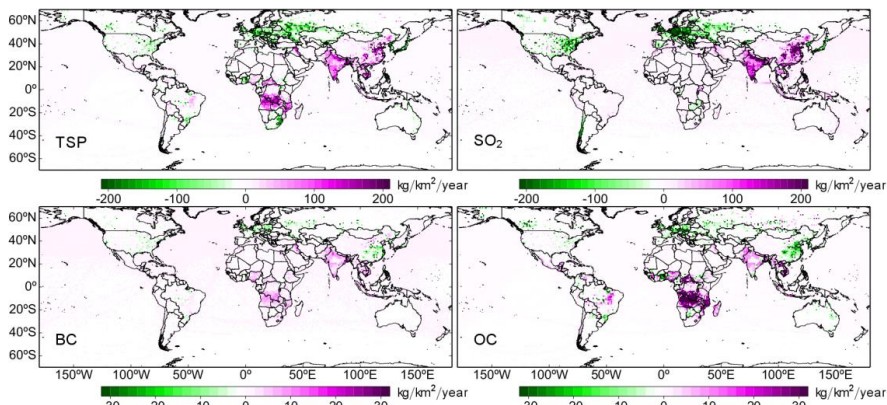

**Figure 10.** Spatial distributions of linear trends (units: kg/km$^2$/year) in total anthropogenic emissions of total suspended particles (TSP), SO$_2$, black carbon (BC), and organic carbon (OC) during 1980–2014 derived from the Peking University emissions inventory (http://inventory.pku.edu.cn/) (Huang et al., 2014). Only linear trend values with statistical significance at the 95% confidence level are shown.



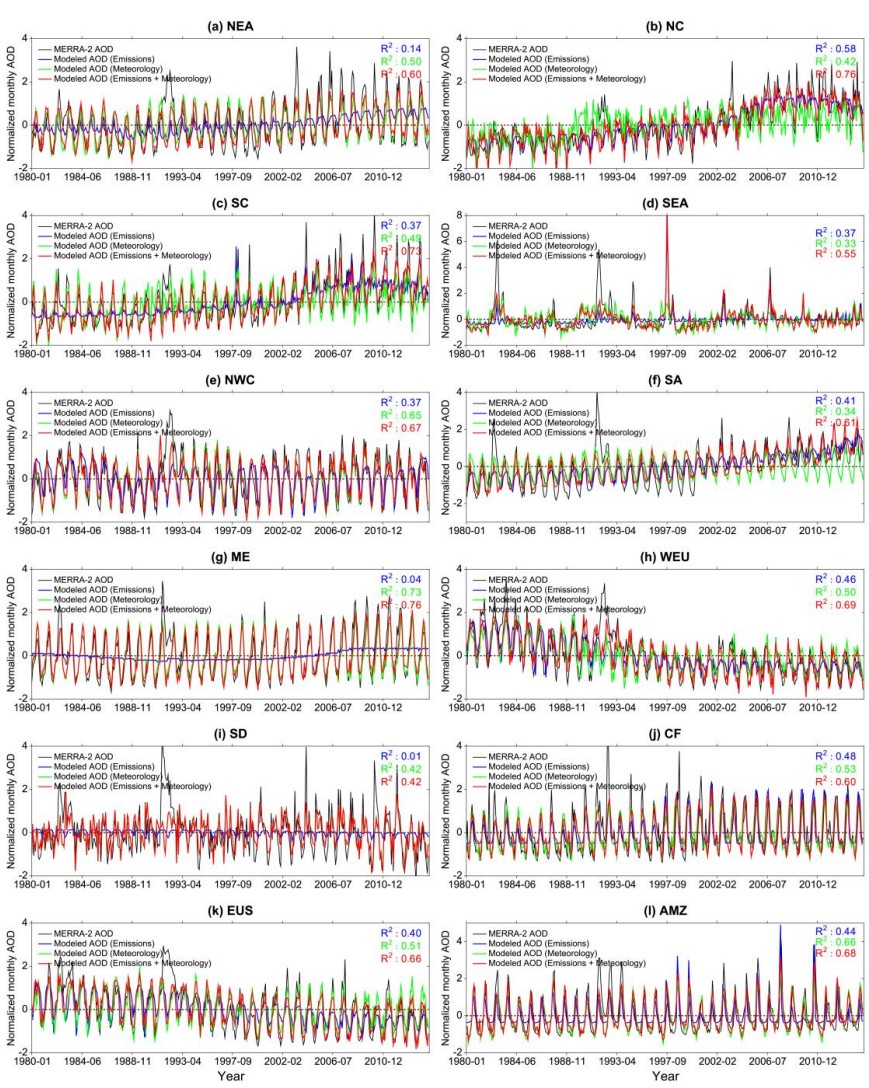

1473

**Figure 11.** Time series of MERRA-2 (in black) and modeled AOD monthly normalized anomalies from 1980 to
2014 over the 12 regions of interest. The coefficient of determination ($R^2$) of the regression fit of the stepwise
MLR model with emission factors (in blue), meteorology (in green), and both emissions and meteorology (in red)
as predictors are given in the top-right of each panel.

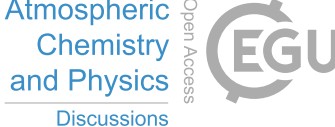



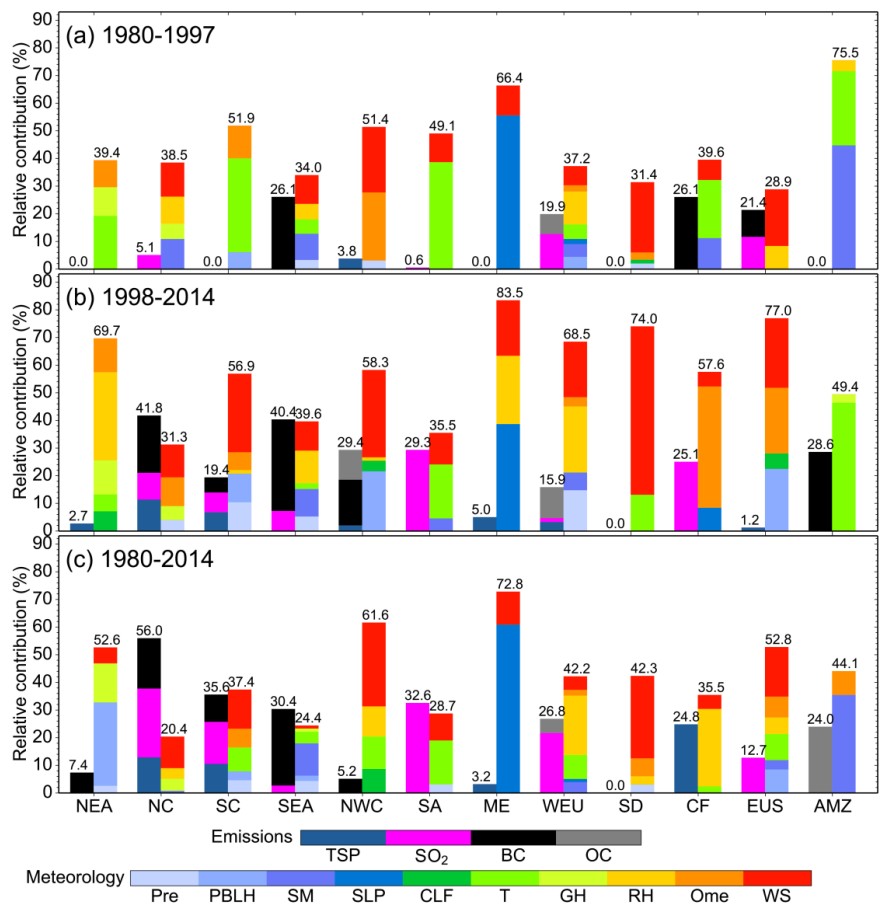

**Figure 12.** The LMG method–estimated relative contributions (%) of total variances in the stepwise MLR model explained by the local emission factors (left-hand bars) and meteorological variables (right-hand bars) over the 12 regions of interest during three periods: (a) 1980–1997 (top panel); (b) 1998–2014 (middle panel); and (c) 1980–2014 (bottom panel). Note that meteorological parameters were combined as follows: temperature, T (Ts, $T_{850}$, $T_{700}$, $T_{500}$, $dT_{900-s}$, $dT_{850-s}$); geopotential height, GH ($GH_{850}$, $GH_{700}$, $GH_{500}$); relative humidity, RH ($RH_s$, $RH_{850}$, $RH_{700}$, $RH_{500}$); vertical velocity, Ome ($Ome_{850}$, $Ome_{700}$, $Ome_{500}$); and wind speed, WS ($U_{850}$, $U_{700}$, $U_{500}$, $V_{850}$, $V_{700}$, $V_{500}$, $WS_s$, $WS_{850}$, $WS_{700}$, $WS_{500}$, $VWS_{500-850}$). Refer to Table S3 for the detailed relative contributions of each variable in the stepwise MLR models.