# Peer review of "Large contribution of meteorological factors to inter-decadal"

_Atmospheric Chemistry and Physics, 2019_

## Referee Comment (RC1) · Anonymous Referee #1 · 22 May 2019

This study makes an extensive study of the global trends of aerosol optical depth during more then 30 years, with an emphasis on the quantitative estimation

of the different weights of emission and meteorological factors in the inter-decadal changes. Different aerosol optical deth sources have been employed, such

as satellite derived (MODIS and MISR), ground based (AERONET and CARSNET) and reanalysis (MERRA-2). 12 Regions of Interest were defined, using more than 400

ground stations. The method employed for the derivation of the different weights is based on a stepwise multiple linear regression model, further completed

with the Lindeman, Merenda and Gold method to quantitatively evaluate the contribution of each driving factor.

The objective of the paper is challenging, implying the use of good quality and large databases. The mthods used are complex and recognised by the

international community. The validation made in the first part of the study important, given that MERRA-2 performance on a global scope was not assessed

before. The results reached are of upmost importance, with adequate references to individual regional results that are complementary and support the

conclusions. The paper is well written, with very few and minor mistakes; no flaws detected on the English grammar or style.

On a first read I would like to make a few comments:

- Abstract, line 32: I would change the "in-situ" measurements with "ground-based measurements", even if it is not incorrect. - Line 129: Earth system? - Lines 180 - 183: The uncertainty estimation is based on Level 2 data, but this analysis uses Level 3 data. PLease consider using Level 3 uncertainty

estimations such as Ruiz-Arias et al. (2013): J. A. Ruiz-Arias, J. Dudhia, C. A. Gueymard, and D. Pozo-Vazquez. Assessment of the Level-3 MODIS daily aerosol

optical depth in the context of surface solar radiation and numerical weather modeling. Atmos. Chem. Phys., 13, 675–692, 2013. doi:10.5194/acp-13-675-2013 - Line 185: earth? - Line 201: I think it should be written "AERONET and CARSNET" - Line 235-236: other independent meteorological fields instead of MERRA-2? - Line 237: etc - Line 343: AERONET - Line 347-348: I don't understand this, based in figure b4. Should be "lower" instead of "greater" than 0.1 and 0.05, respectively? - Line 356: there clear? - Lines 488-490: any reason to explain the change? - Lines 824: the term "wind speed" was used in the previous paragraph, but here WS is used instead. Perhaps using wind speed would be fine, given that the

text includes many acronyms.

---

## Referee Comment (RC2) · Anonymous Referee #2 · 18 Jun 2019

The paper "Large contribution of meteorological factors to inter-decadal changes in regional aerosol optical depth" presents and discusses the long-terms global trends in AOD (1980–2016), based on MERRA-2, MODIS-Terra, MISR and Statistical Models on emission factors and meteorological parameters, driving the inter-decadal AOD changes. The study falls within the scope of ACP. The manuscript is well-written and structured, the presentation clear, the language fluent and the quality of the figures high. The authors have done a thorough job and the results support the conclusions. I recommend publication in ACP, however I recommend the following revisions before it can proceed to be published.

Comments:

1) The manuscript validates MERRA-2 against AERONET, MODIS-Terra and MISR. However the AERONET, MODIS-Terra and MISR datasets are used in the assimilation of MERRA-2, and therefore are not independent. From the used datasets only CARSNET is independent. Therefore, it is suggested to the authors to provide separately the comparison and discussion of MERRA-2 against CARSNET, and modify the manuscript in order to discuss the use of AERONET, MODIS-Terra and MISR only in terms of evaluation and not validation.

2) Page 2, lines 32: AERONET is not in-situ. Please rephrase.

3) Page 4, lines 130-134: The authors are argued to use MODIS-Aqua also in the analysis, despite the two years of less observations. MODIS-Terra has been documented to suffer from degradation issues, therefore the observed trends may be an artifact. In addition, please comment on the possible sensor effect on the presented results of the manuscript.

4) I think it would be beneficial for the manuscript to include a flowchart showing the methodology of the comparison followed by the authors. The entire process can be summarized there along with the methodology requirements followed e.g. the spatial - temporal constraints, screening requirements, wavelength, etc. The information exists in the manuscript but I feel like it is scattered among the sections. Furthermore, I suggest the authors to provide the collocation criteria (both spatial and temporal), the wavelength of AOD studied, since the datasets are very different. For instance how the analysis of MERRA-2 is gridded in 1x1 deg2 resolution, why three-hourly MERRA-2 are compared to hourly AERONET and not three-hourly AERONET and how the dataset is gridded to ROIs? All these information should be gathered in a flow-chart.

5) Comparison methods: It would be interesting to show the Fractional Bias and the Fractional gross error. In additional, please include among the other equations the correlation coefficient equation.

6) Page 8, lines: 295-297: what about the cases of 0.05<P<0.1?

7) Please include a table with as many rows as the number of ROIs to summarize and show simultaneously the following: MAE, RMSE, R, VIF, positive/negative trends (periods 1,2,3), and statistical significance (yes, no) at the 0.05 level.

8) The discussed methodology: How much the selected methodology affect the final trends (e.g. selected periods 1,2,3, boundaries of ROIs, collocated criteria…)?

9) The scientific methods and assumptions are valid, and the authors give proper credit to related work, especially in the introduction and methodology. In the "Results Section" though frequently results on AOD trends are presented, without any explanation/discussion on the observed trends and without the associated references on the discussion and the explanation on the results. The undervalue of the present work is not to simple provide the trends, but to comment on the physical and anthropogenic causes that the trends are associated with. Hereinafter, the authors will find indicative parts of the manuscript to improve by extending the associated parts with discussion and/or references, which I missing when reading the manuscript:

- Page 10, lines 353-355: "which indicates that MERRA-2 overestimates the AOD in these regions"? Where is the overestimation attributed to? (explanation and references)
- Page 12, lines 446-458: ""... global distribution of AOD also shows obvious seasonal differences, with global aerosol loading reaching its maximum in springand summer." (explanation and references)
- Page 12, lines 449-458: References missing.
- Page 13, lines 465-469: References missing.
- Page 13, lines 487-492: Explanations and references missing.
- Page 14, lines 524-530: Explanations and references missing.
- Page 15, lines 546: "except for the positive trend that still existed in the marine area of the Southern Hemisphere, the fluctuations inother regionswere smaller and relatively stable.": Explanations and references missing.
- Page 15, lines 549-552: Explanation missing.
- Page 15, lines 554-556: Explanations and references missing.
- Page 15, lines 557-558: Explanations and references missing.
- Page 15, lines 559-561: Explanations and references missing.
- Page 15, lines 561-564: Explanations and references missing.
- Page 16, lines 604-605: Explanations and references missing.
- Page 16, lines 607-619: The entire paragraph needs improvement. Results are presesented, not explained. Explanations and references missing.
- Page 16, lines 629: Explanations and references missing.
- Page 16, lines 632: Explanations and references missing.
- Page 17, lines 634: References missing.
- Page 17, lines 634-635: References missing.
- Page 17, lines 635-637: References missing.
- Page 17, lines 638-640: References missing.
- Page 17, lines 640-643: References missing.
- Page 17, lines 645-646: Explanations and references missing.
- Page 17, lines 653-656: References missing.
- Page 17, lines 658-661: References missing.
- Page 17, lines 662-663: Explanations and references missing.
- Page 17, lines 667: Explanation missing.
- Page 17, lines 671: References missing.
- Page 17, lines 672-675: References missing.
- Page 18, lines 678-680: Explanations and references missing.
- Page 18, lines 683-684: References missing.
- Page 18, lines 687: References missing.
- Page 18, lines 709-710: Explanations and references missing.

- Page 18, lines 710-713: Explanations and references missing.
- Page 19, lines 750-755: References missing.

---

## Author Comment (AC2) · 17 Jul 2019

The following is a point-to-point response to the reviewer's comments. We have studied comments carefully and have made correction which we hope meet with approval. Revised portion are marked in red in the revised paper.

**Reviewer #2**

The paper "Large contribution of meteorological factors to inter-decadal changes in regional aerosol optical depth" presents and discusses the long-terms global trends in AOD (1980–2016), based on MERRA-2, MODIS-Terra, MISR and Statistical Models on emission factors and meteorological parameters, driving the inter-decadal AOD changes. The study falls within the scope of ACP. The manuscript is well-written and structured, the presentation clear, the language fluent and the quality of the figures high. The authors have done a thorough job and the results support the conclusions. I recommend publication in ACP, however I recommend the following revisions before it can proceed to be published.

**Response:** Thank you for your positive comments on our article. We have revised it in accordance with your comments or suggestions. For detailed revisions, please refer to the following sections

**Comments**

1. The manuscript validates MERRA-2 against AERONET, MODIS-Terra and MISR. However the AERONET, MODIS-Terra and MISR datasets are used in the assimilation of MERRA-2, and therefore are not independent. From the used datasets only CARSNET is independent. Therefore, it is suggested to the authors to provide separately the comparison and discussion of MERRA-2 against CARSNET, and modify the manuscript in order to discuss the use of AERONET, MODIS-Terra and MISR only in terms of evaluation and not validation

**Response:** Thanks for your thoughtful suggestion. We have made the following revisions to section 3.1.

(1). As you suggested, considering that AERONET has been assimilated into the MERRA-2 system, CARSNET does not. Therefore, we made two independent comparisons in Section 3.1 in the revised paper (MERRA-2 *versus* AERONET and MERRA-2 *versus* CARSNET). Detailed revisions can be found in the revised paper. In addition, the previous figure 2 has been separated into two separate comparison graphs for better comparison, and the revised figure 3 is shown below. Note that we have added the three additional statistical metrics (i.e. MFE, FGE and IOA) as recommended by the reviewers.

(2). **Abstract Section:** The sentence: "Evaluation of the MERRA-2 AOD with the combined in-situ measurements of AERONET and the China Aerosol Remote Sensing Network indicated significant spatial agreement on the global scale ($r = 0.84$, RMSE $= 0.14$, and MAE $= 0.07$)." has been changed to "Evaluation of the MERRA-2 AOD with the ground-based measurements of AERONET indicated significant spatial agreement on the global scale ($r = 0.85$, RMSE $= 0.12$, MFE$= 38.7\%$, FGE $= 9.86\%$, and IOA$= 0.94$). However, when AOD observations from the China Aerosol Remote Sensing Network (CARSNET) were employed for independent verification, the

results showed that MERRA-2 AODs generally underestimated CARSNET AODs in China (RMB =0.72 and FGE=−34.3%).".

[Figure]

**Figure 3.** Evaluation of the three-hourly MERRA-2 AOD against the (a) AERONET and (b) CARSNET AODs. The color-coded dots indicate the number of samples. The solid red line is the line of best fit and the black dashed line is the 1:1 line. For descriptions of statistical metrics, see the comparison methods section.

2. Page 2, lines 32: AERONET is not in-situ. Please rephrase

**Response:** According to the reviewer's good suggestions. "in-situ" has been changed to "ground-based".

3. Page 4, lines 130-134: The authors are argued to use MODIS-Aqua also in the analysis, despite the two years of less observations. MODIS-Terra has been documented to suffer from degradation issues, therefore the observed trends may be an artifact. In addition, please comment on the possible sensor effect on the presented results of the manuscript

**Response:** It should be pointed out that there are two reasons why we chose MODIS/Terra in this study. First, in order to ensure the consistency of the time length of three AOD datasets, MODIS/Aqua observed in less than two years was not selected. Secondly, and most importantly, we have assessed the global trends of MODIS/Aqua and MODIS/Terra AOD during overlapping periods (2003-2016) (see Figure S1 below). Compared with MODIS/Aqua AOD trend, MODIS/Terra AOD shows similar performance worldwide (including spatial-temporal consistency and distribution patterns of trend values). Combining the above two reasons, MODIS/Terra was used in this study.

In addition, we agree with the reviewer's reference to the degradation issues of the MODIS-Terra. However, due to the degradation of MODIS sensors, a new calibration approach in the latest version of C6 has been used in order to remove major non-polarimetric calibration trends from the MODIS data (Levy et al., 2013, 2015; Lyapustin et al., 2014). Therefore, MODIS/Terra can show similar performance to MODIS/Aqua, as you can see in Fig.S1. In the revised paper, we also re-describe this part as follows:

"In addition, compared with the linear trend in MODIS/Aqua AOD during 2003-2016, MODIS/Terra AOD shows similar performance worldwide (including spatial-temporal consistency and distribution patterns of trend values) (Fig. S1), although the Terra sensor has been documented to suffer from degradation issues. The similar performance between MODIS/Terra and MODIS/Aqua is mainly attributed to a new calibration approach in the C6 version, which can remove major non-polarimetric calibration trends from the MODIS data (Levy et al., 2013, 2015; De Leeuw et al., 2018)."

[Figure]

**Fig. S1.** Spatial distributions of annual trends in AOD calculated from the time series value of the de-seasonalized monthly anomaly from (a) MODIS/Terra and (b) MODIS/Aqua between 2003 and 2016. The grid areas with black dots indicate statistical significance at the 95% confidence level.

**Reference:**
1. Levy RC, Mattoo S,Munchak LA, Remer LA, Sayer AM, Patadia F, Hsu NC (2013) The Collection 6 MODIS aerosol products over land and ocean. AtmosMeas Tech 6(11):2989–3034. https://doi.org/10.5194/ amt-6-2989-2013
2. Levy RC, Munchak LA, Mattoo S, Patadia F, Remer LA, Holz RE (2015) Towards a long-term global aerosol optical depth record: applying a consistent aerosol retrieval algorithm to MODIS and VIIRS- observed reflectance. Atmos Meas Tech 8(10):4083–4110. https:// doi.org/10.5194/amt-8-4083-2015

3. De Leeuw, G., Sogacheva, L., Rodriguez, E., Kourtidis, K., Georgoulias, A. K., Alexandri, G., Amiridis, V., Proestakis, E., Marinou, E., Xue, Y. and Van Der A, R.: Two decades of satellite observations of AOD over mainland China using ATSR-2, AATSR and MODIS/Terra: Data set evaluation and large-scale patterns, Atmos. Chem. Phys., 18(3), 1573–1592, doi:10.5194/acp-18-1573-2018, 2018.

4. I think it would be beneficial for the manuscript to include a flowchart showing the methodology of the comparison followed by the authors. The entire process can be summarized there along with the methodology requirements followed e.g. the spatial - temporal constraints, screening requirements, wavelength, etc. The information exists in the manuscript but I feel like it is scattered among the sections. Furthermore, I suggest the authors to provide the collocation criteria (both spatial and temporal), the wavelength of AOD studied, since the datasets are very different. For instance how the analysis of MERRA-2 is gridded in 1x1 deg2 resolution, why three-hourly MERRA-2 are compared to hourly AERONET and not three-hourly AERONET and how the dataset is gridded to ROIs? All these information should be gathered in a flow-chart

**Response:** According to the reviewer's good suggestions. We have integrated the comparison methods, trend estimated and factor contribution into the following flowchart. We believe that this flowchart can deepen the reader's understanding of the structure of this paper.

[Figure]

Figure 2. Flowchart with the procedure followed for (a) the evaluation of MERRA-2 global AOD using the AERONET and CARSNET ground-based reference dataset, and (b) the evaluation of global and regional AOD trends.

5. Comparison methods: It would be interesting to show the Fractional Bias and the Fractional gross error. In additional, please include among the other equations the correlation coefficient equation.

**Response:** According to the reviewer's good suggestions. We have added the two statistical metrics (i.e. MFE and FGE) recommended by the reviewers to the revised

paper. In addition, an additional metric (IOA, the index of agreement) was also considered. The three newly added metrics, along with the equation for the correlation coefficient (R), are updated simultaneously into the revised paper. Please refer to the revised paper for specific revisions.

At the same time, these statistical metrics have also been updated in Figures 3, 4 S2, and S3. The updated figures are shown below.

[Figure]

**Figure 3.** Evaluation of the three-hourly MERRA-2 AOD against the (a) AERONET and (b) CARSNET AODs. The color-coded dots indicate the number of samples. The solid red line is the line of best fit and the black dashed line is the 1:1 line. For descriptions of statistical metrics, see the comparison methods section.

[Figure]

**Figure 4**. Comparison of the three-hourly MERRA-2 AOD datasets with AOD observations of 468 AERONET sites worldwide and 37 CARSNET sites in China: site performance maps for the (a) correlation coefficient (*R*), (b) mean absolute error (MAE), root-mean-square error (RMSE), (c) relative mean bias (RMB), (d) mean fractional error (MFE), (e) fractional gross error (FGE), and (f) the index of agreement (IOA) between MERRA-2 AOD and

ground-based AOD observations. The size of the circles in Fig.4b represents the RMSE and their inner color represents the MAE. The bars in the lower left inset in each panel represent the frequency distribution histograms for the *R*, MAE, RMSE, RMB, MFE, FGE and IOA between MERRA-2 and all ground-based observations incorporating AERONET and CARSNET, respectively. Note that all sites within each region of interest (ROI) are integrated to assess the accuracy of the MERRA-2 AOD dataset in that area. The performance of the MERRA-2 AOD dataset in each ROI is illustrated in Figs. S2 and S3.

[Figure]

**Fig. S2.** Validations of AERONET AOD measurements against the three-hourly MERRA-2 AOD over the 12 ROIs, as defined in Figure 1. The color-coded dots indicate the number of samples. Where the solid red line is the line of best fit and the black dashed line is the 1:1 line.

[Figure]

**Fig. S3.** Validations of CARSNET AOD measurements against the three-hourly MERRA-2 AOD over (a) NC, (b) SC, and (c) NWC, as defined in Figure 1. The color-coded dots indicate the number of samples. Where the solid red line is the line of best fit and the black dashed line is the 1:1 line.

6.  Page 8, lines: 295-297: what about the cases of 0.05<P<0.1?

**Response:** We use P < 0.1 instead of P < 0.05 when removing explanatory variables in the stepwise MLR model. The main reasons are as follows:

In each step of the MLR model, the model selects the most powerful and significant (p<0.05) predictor explaining the residual. As the number of explanatory variables in the model increases, the explanatory power of the model increases, but if P<0.05 is used to remove the variables, the variables retained in the final model will be very limited. In addition, in order to eliminate the over-fitting problem caused by multivariate collinearity in our model, this study also further utilized VIF to filter out those variables that have significant collinearity. Methods that use the same threshold (i.e. p<0.05 for the selection step and P<0.1 for the remove step) are also used in other studies, such as Lu et al. (2016) and Zhai et al. (2019) below.

**Reference:**

1.  Lu, X., Zhang, L., Yue, X., Zhang, J., Jaffe, D. A., Stohl, A., Zhao, Y. and Shao, J.: Wildfire influences on the variability and trend of summer surface ozone in the mountainous western United States, Atmos. Chem. Phys., doi:10.5194/acp-16-14687-2016, 2016.
2.  Zhai, S., Jacob, D. J., Wang, X., Shen, L., Li, K., Zhang, Y., Gui, K., Zhao, T., and Liao, H.: Fine particulate matter (PM2.5) trends in China, 2013–2018: contributions from meteorology, Atmos. Chem. Phys. Discuss., https://doi.org/10.5194/acp-2019-279, in review, 2019.

7.  Please include a table with as many rows as the number of ROIs to summarize and show simultaneously the following: MAE, RMSE, R, VIF, positive/negative trends (periods 1,2,3), and statistical significance (yes, no) at the 0.05 level.

**Response:** According to the reviewer's good suggestions. We have added a table (including R, MAE, RMSE, RMB, MFE, GFE, and IOA) to summarize the performance of MERRA-2 AOD in different the 12 ROIs. What's more, a summary of some of the other parameters (i.e. positive/negative trends (periods 1, 2, and 3), and statistical significance (yes, no) at the 0.05 level.) you mentioned can be found in Table S1.

Table 2. Statistical measures of the hourly AERONET and CARSNET AODs versus MERRA-2 AOD over the 12 regions of interest.

| ROIs | Number of sites | Number of collocations | R | MAE | RMSE | RMB | MFE (%) | FGE (%) | IOA |
|------|-----------------|------------------------|------|------|------|------|---------|---------|------|
| NEA | 13 | 35066 | 0.79 | 0.10 | 0.16 | 0.93 | 33.18 | -2.65 | 0.92 |
| NC | 3 | 16782 | 0.80 | 0.25 | 0.42 | 0.71 | 45.44 | -23.85 | 0.78 |
| SC | 2 | 3616 | 0.87 | 0.08 | 0.13 | 1.01 | 24.73 | 5.25 | 0.95 |
| SEA | 17 | 32112 | 0.79 | 0.12 | 0.24 | 0.84 | 31.26 | -8.52 | 0.86 |
| NWC | 1 | 4633 | 0.85 | 0.03 | 0.05 | 1.01 | 30.74 | 1.98 | 0.98 |
| SA | 13 | 33385 | 0.84 | 0.11 | 0.18 | 0.87 | 34.54 | -8.06 | 0.93 |
| ME | 10 | 34312 | 0.95 | 0.04 | 0.07 | 1.02 | 12.89 | 4.13 | 0.98 |
| WEU | 81 | 252767 | 0.79 | 0.04 | 0.07 | 0.95 | 32.91 | 2.01 | 0.97 |
| SD | 14 | 69982 | 0.81 | 0.14 | 0.20 | 0.97 | 33.22 | 4.40 | 0.91 |
| CF | 5 | 12380 | 0.83 | 0.08 | 0.14 | 0.75 | 35.78 | -22.96 | 0.93 |
| EUS | 38 | 105577 | 0.70 | 0.07 | 0.11 | 1.11 | 42.28 | 17.82 | 0.94 |
| AMZ | 8 | 21105 | 0.82 | 0.08 | 0.19 | 0.84 | 35.84 | -1.73 | 0.89 |
| NC[a] | 12 | 27508 | 0.70 | 0.23 | 0.33 | 0.71 | 47.31 | -35.45 | 0.81 |
| SC[a] | 2 | 2346 | 0.74 | 0.15 | 0.21 | 0.92 | 30.85 | -8.01 | 0.90 |
| NWC[a] | 3 | 10103 | 0.67 | 0.20 | 0.33 | 0.69 | 45.17 | -26.00 | 0.78 |

[a] indicates the statistical results for CARSNET sites.

8. The discussed methodology: How much the selected methodology affect the final trends (e.g. selected periods 1, 2, 3, boundaries of ROIs, collocated criteria…)?

**Response:** There is no doubt that different selection methods (such as the selected period, the selected boundaries of ROIs, etc.) will have different effects on the final trends when conducting regional trend assessments. Nevertheless, the method chosen for the regional trend assessment of AOD in this study is scientific and effective.

**Regarding the selections of ROIs and their boundaries**, the 12 ROIs selected in this study have experienced substantial natural or anthropogenic aerosol pollution and received considerable attention in other aerosol climate studies (such as Zhao et al., 2018; Chin et al., 2014; Klingmüller et al., 2016; Hsu et al., 2012; Proestakis et al., 2018; Lee et al., 2016; De Leeuw et al., 2018). In addition, in order to better compare with the results of previous studies, the selection of regional boundaries is as consistent as possible with previous studies.

**Regarding the selection of periods**, the three selected periods (i.e. periods 1, 2, and 3) not only covered the entire study period (period 1), but also included two independent periods (periods 2 and 3) with similar time lengths. A comparative assessment of trends over the three periods can deepen our understanding of the evolution of global and regional AOD trends. In addition, in order to solve the incomplete understanding of regional long-term evolution trend in AOD caused by the choice of a fixed research period, this study also systematically evaluates the evolution process of AOD trend in different regions using sliding-window trend analysis method.

**Reference:**

1. Chin, M., Diehl, T., Tan, Q., Prospero, J. M., Kahn, R. A., Remer, L. A., Yu, H., Sayer, A. M., Bian, H., Geogdzhayev, I. V., Holben, B. N., Howell, S. G., Huebert, B. J., Hsu, N. C., Kim, D., Kucsera, T. L., Levy, R. C., Mishchenko, M. I., Pan, X., Quinn, P. K., Schuster, G. L., Streets, D. G., Strode, S. A. and Torres, O.: Multi-decadal aerosol variations from 1980 to 2009: A perspective from observations and a global model, Atmos. Chem. Phys., 14(7), 3657–3690, doi:10.5194/acp-14-3657-2014, 2014.

2. Hsu, N. C., Gautam, R., Sayer, A. M., Bettenhausen, C., Li, C., Jeong, M. J., Tsay, S. C. and Holben, B. N.: Global and regional trends of aerosol optical depth over land and ocean using SeaWiFS measurements from 1997 to 2010, Atmos. Chem. Phys., 12(17), 8037–8053, doi:10.5194/acp-12-8037-2012, 2012.

3. Klingmüller, K., Pozzer, A., Metzger, S., Stenchikov, G. L. and Lelieveld, J.: Aerosol optical depth trend over the Middle East, Atmos. Chem. Phys., 16(8), 5063–5073, doi:10.5194/acp-16-5063-2016, 2016.

4. Lee, H., Kalashnikova, O. V., Suzuki, K., Braverman, A., Garay, M. J. and Kahn, R. A.: Climatology of the aerosol optical depth by components from the Multi-angle Imaging SpectroRadiometer (MISR) and chemistry transport models, Atmos. Chem. Phys., 16(10), 6627–6640, doi:10.5194/acp-16-6627-2016, 2016.

5. De Leeuw, G., Sogacheva, L., Rodriguez, E., Kourtidis, K., Georgoulias, A. K., Alexandri, G., Amiridis, V., Proestakis, E., Marinou, E., Xue, Y. and Van Der A, R.: Two decades of satellite observations of AOD over mainland China using ATSR-2, AATSR and MODIS/Terra: Data set evaluation and large-scale patterns, Atmos. Chem. Phys., 18(3),

1573–1592, doi:10.5194/acp-18-1573-2018, 2018.

6.  Proestakis, E., Amiridis, V., Marinou, E., Georgoulias, A. K., Solomos, S., Kazadzis, S., Chimot, J., Che, H., Alexandri, G., Binietoglou, I., Daskalopoulou, V., Kourtidis, K. A., De Leeuw, G. and Van Der A, R. J.: Nine-year spatial and temporal evolution of desert dust aerosols over South and East Asia as revealed by CALIOP, Atmos. Chem. Phys., 18(2), 1337–1362, doi:10.5194/acp-18-1337-2018, 2018.

7.  Zhao, B., Jiang, J. H., Diner, D. J., Su, H., Gu, Y., Liou, K.-N., Jiang, Z., Huang, L., Takano, Y., Fan, X. and Omar, A. H.: Intra-annual variations of regional aerosol optical depth, vertical distribution, and particle types from multiple satellite and ground-based observational datasets, Atmos. Chem. Phys. Discuss., 2018, 1–33, doi:10.5194/acp-2018-110, 2018.

9.  The scientific methods and assumptions are valid, and the authors give proper credit to related work, especially in the introduction and methodology. In the "Results Section" though frequently results on AOD trends are presented, without any explanation/discussion on the observed trends and without the associated references on the discussion and the explanation on the results. The undervalue of the present work is not to simple provide the trends, but to comment on the physical and anthropogenic causes that the trends are associated with. Hereinafter, the authors will find indicative parts of the manuscript to improve by extending the associated parts with discussion and/or references, which I missing when reading the manuscript:

**Response:** Thanks for your thoughtful suggestion. We have explained these sentences appropriately and added references as required by the reviewers. At the same time, it should be pointed out that for the regional trends that appear in Section 3.4, we have partially explained these trends according to emissions and meteorology in Section 3.5 of the original manuscript.

10. Page 10, lines 353-355: "which indicates that MERRA-2 overestimates the AOD in these regions"? Where is the overestimation attributed to? (explanation and references)

**Response:** Thanks for pointing this out. We have added some appropriate explanations and reference for this sentence, as shown below:

"This overestimation may be attributed to the bias of MISR AOD in these areas (not shown here) and the fact that AERONET was not assimilated in MERRA-2 until 1999 (Buchard et al., 2017)."

**Reference:**
1.  Buchard, V., Randles, C. A., da Silva, A. M., Darmenov, A., Colarco, P. R., Govindaraju, R., Ferrare, R., Hair, J., Beyersdorf, A. J., Ziemba, L. D. and Yu, H.: The MERRA-2 aerosol reanalysis, 1980 onward. Part II: Evaluation and case studies, J. Clim., 30(17), 6851–6872, doi:10.1175/JCLI-D-16-0613.1, 2017.

11. Page 12, lines 446-458: ""... global distribution of AOD also shows obvious seasonal differences, with global aerosol loading reaching its maximum in spring and summer." (explanation and references)

**Response:** Thanks for your thoughtful suggestion. In fact, we have made a reasonable explanation for this sentence in the original draft. We explain this phenomenon from two aspects. The explanation in the original text is as follows: "On the one hand, this can mainly be attributed to….. On the other hand…."

12. Additional explanations and/or references to the needs identified by reviewers
**Response:** For most of the missing references and explanations mentioned by reviewers, we have provided them as much as possible. For specific revisions, please refer to the revised version.

---

## Author Comment (AC1)

The following is a point-to-point response to the reviewer's comments. We have studied comments carefully and have made correction which we hope meet with approval. Revised portion are marked in red in the revised paper.

**Reviewer #1**
This study makes an extensive study of the global trends of aerosol optical depth during more than 30 years, with an emphasis on the quantitative estimation of the different weights of emission and meteorological factors in the inter-decadal changes. Different aerosol optical deth sources have been employed, such as satellite derived (MODIS and MISR), ground based (AERONET and CARSNET) and reanalysis (MERRA-2). 12 Regions of Interest were defined, using more than 400 ground stations. The method employed for the derivation of the different weights is based on a stepwise multiple linear regression model, further completed with the Lindeman, Merenda and Gold method to quantitatively evaluate the contribution of each driving factor. The objective of the paper is challenging, implying the use of good quality and large databases. The mthods used are complex and recognised by the international community. The validation made in the first part of the study important, given that MERRA-2 performance on a global scope was not assessed before. The results reached are of upmost importance, with adequate references to individual regional results that are complementary and support the conclusions. The paper is well written, with very few and minor mistakes; no flaws detected on the English grammar or style.
**Response:** Thank you for your positive comments on our article. We have revised it in accordance with your comments or suggestions. For detailed revisions, please refer to the following sections

**Comments**
1. Abstract, line 32: I would change the "in-situ" measurements with "ground-based measurements", even if it is not incorrect.
**Response:** According to the reviewer's good suggestions. The word "in-situ" has been changed to "ground-based".

2. Line 129: Earth system?
**Response:** According to the reviewer's good suggestions. The word "the Earth system" has been changed to "the Earth-atmosphere system".

3. Lines 180 -183: The uncertainty estimation is based on Level 2 data, but this analysis uses Level 3 data. PLease consider using Level 3 uncertainty estimations such as Ruiz-Arias et al. (2013): J. A. Ruiz-Arias, J. Dudhia, C. A. Gueymard, and D. Pozo-Vazquez. Assessment of the Level-3 MODIS daily aerosol optical depth in the context of surface solar radiation and numerical weather modeling. Atmos. Chem. Phys., 13, 675–692, 2013. doi:10.5194/acp-13-675-2013
**Response:** Thanks for pointing this out. Unfortunately, we did not use the reference mentioned by the reviewer because the C5.1 version was evaluated in this paper,

while the C6.1 version was used in our study. Here, we refer to an assessment of the latest C6.1 L3 data. We have described the accuracy of L3 and added references. Detailed revisions are as follows:

"The average MAE (RMSE) of the Level 3 MODIS/Terra DTB monthly AOD data have been estimated to be about 0.075 (0.120) over land (Wei et al., 2019)."

**Reference:**
1. Wei, J., Peng, Y., Guo, J. and Sun, L.: Performance of MODIS Collection 6.1 Level 3 aerosol products in spatial- temporal variations over land, Atmos. Environ., 206, 30–44, doi:10.1016/j.atmosenv.2019.03.001, 2019

**4.** Line 185: earth?

**Response:** According to the reviewer's good suggestions. The word "the Earth atmosphere" has been changed to "the Earth's atmosphere".

**5.** Line 201: I think it should be written "AERONET and CARSNET"
**Response:** Corrected

**6.** Line 235-236: other independent meteorological fields instead of MERRA-2?
**Response:** First, MERRA-2 meteorological field has similar or even better performance than other reanalysis data, because it is the first reanalysis dataset that assimilates aerosol satellite observation data and considers aerosol radiation feedback. Secondly, considering the difference of spatial resolution between MERRA-2 AOD and other reanalysis meteorological fields (such as NECP, JRA, etc.), if other reanalysis meteorological fields are used, the conclusion may be affected. Therefore, we believe that the MERRA-2 meteorological field is suitable for this study.

**7.** Line 237: etc
**Response:** Corrected

**8.** Line 343: AERONET
**Response:** Corrected

**9.** Line 347-348: I don't understand this, based in figure b4. Should be "lower" instead of "greater" than 0.1 and 0.05, respectively?
**Response:** Thanks for pointing this out. The word "greater" has been changed to "lower".

**10.** Line 356: there clear?
**Response:** Thanks for your thoughtful suggestion. The word "Europe" has been changed to "southern Europe"

**11.** Lines 488-490: any reason to explain the change?

**Response:** Thanks for your thoughtful suggestion. We have added some appropriate explanations and reference for this sentence, as shown below:

"This shift may be attributed to the fact that MERRA-2 did not assimilate any land-based AOD observations before 1999, which made it difficult for the model to simulate the monthly variation of regional AOD (Gelaro et al., 2017; Buchard et al., 2017)."

**Reference:**
1. Gelaro, R., McCarty, W., Suárez, M. J., Todling, R., Molod, A., Takacs, L., Randles, C. A., Darmenov, A., Bosilovich, M. G., Reichle, R., Wargan, K., Coy, L., Cullather, R., Draper, C., Akella, S., Buchard, V., Conaty, A., da Silva, A. M., Gu, W., Kim, G. K., Koster, R., Lucchesi, R., Merkova, D., Nielsen, J. E., Partyka, G., Pawson, S., Putman, W., Rienecker, M., Schubert, S. D., Sienkiewicz, M. and Zhao, B.: The modern-era retrospective analysis for research and applications, version 2 (MERRA-2), J. Clim., 30(14), 5419–5454, doi:10.1175/JCLI-D-16-0758.1, 2017.
2. Buchard, V., Randles, C. A., da Silva, A. M., Darmenov, A., Colarco, P. R., Govindaraju, R., Ferrare, R., Hair, J., Beyersdorf, A. J., Ziemba, L. D. and Yu, H.: The MERRA-2 aerosol reanalysis, 1980 onward. Part II: Evaluation and case studies, J. Clim., 30(17), 6851–6872, doi:10.1175/JCLI-D-16-0613.1, 2017.

**12.** Lines 824: the term "wind speed" was used in the previous paragraph but here WS is used instead. Perhaps using wind speed would be fine, given that the text includes many acronyms

**Response:** Corrected.

---

## Author Response (AR2)

The following is a point-to-point response to the editor's comments. We have studied comments carefully and have made correction which we hope meet with approval. Revised portion are marked in red in the revised paper.

The paper is an important contribution to the aerosol community.
**Response:** Thank you for your positive comments on our article. We have revised it in accordance with your comments or suggestions. For detailed revisions, please refer to the following sections

**Comments**
1. I think it has to be mentioned in the case that of Merra-2 and aeronet comparison that the data sets are not independent. I agree that "it is difficult to disentangle the influence of each assimilated data set alone on the overall accuracy of MERRA 2" but still the above basic feature has to be mentioned.

**Response:** According to the editor's good suggestions. We have added some appropriate descriptions for this basic feature, as shown below:

"Strictly speaking, we need to point out that MERRA-2 and AERONET are not independent of each other (after 1999). Nevertheless, we hope that this assessment will still provide some reference for other studies using the MERRA-2 AOD dataset."

2. Not all references mentioned in the authors answer to the reviewers have been included in the final manuscript.

**Response:** All references have been added to the revised version.

3. A few lines about the Aeronet, MODIS, Merra-2 uncertainty has to be included with a comment with their effect on the trend and AOD statistics.

**Response:** According to the editor's good suggestions. We have added some comments on the effect of uncertainties in different data sets on trend assessment results, as shown below:

"It should be noted that different data sets may have a certain effect on global and regional trend assessment due to their own uncertainties. Nevertheless, we include them for completeness but exercise with caution when interpreting the differences in trend values between different data sets."